# Near-Surface Snow Particle Dynamics from Particle Tracking Velocimetry and Turbulence Measurements during Alpine Blowing Snow Storms

Nikolas O. Aksamit[1], John W. Pomeroy

[1]Centre for Hydrology, University of Saskatchewan, Saskatoon, S7N 5C8, Canada

*Correspondence to*: Nikolas O. Aksamit (n.aksamit@usask.ca)

**Abstract.** Many blowing snow conceptual and predictive models have been based on simplified two-phase flow dynamics derived from time-averaged observations of bulk flow conditions in blowing snow storms. Measurements from the first outdoor application of Particle Tracking Velocimetry (PTV) of near-surface blowing snow yields new information on mechanisms for blowing snow initiation, entrainment, and rebound, whilst also confirming some findings from wind tunnel observations. Blowing snow particle movement is influenced by complex surface flow dynamics, including saltation development from creep that has not previously been measured for snow. Comparisons with 3D atmospheric turbulence measurements show that blowing snow particle motion immediately above the snow surface responds strongly to high frequency turbulent motions. Momentum exchange from wind to the dense near-surface particle-laden flow appears significant and makes an important contribution to blowing snow mass flux and saltation initiation dynamics. The more complete and accurate description of near-surface snow particle motions observable using PTV may prove useful for improving blowing snow model realism and accuracy.

## 1 Introduction

Wind transport of snow influences the variability of alpine summer runoff (*Pomeroy et al.*, 2012; *Winstral et al.,* 2013), is a large contributor to the growth or ablation of small mountain glaciers (*Dyunin and Kotlyakov*, 1980), and can contribute snow loading to avalanche prone areas (*Schweizer et al*., 2003). Time-averaged blowing snow field measurements often present an oversimplified view of a highly variable and unsteady natural phenomenon. Physical snow trap mechanisms only provide mass flux averages over prolonged collection periods (*Budd et al.*, 1966). Snow particle counters only

recently began providing point measurements of particle speed (*Nishimura et al.,* 2014) along with particle size and number flux values (*Schmidt*, 1984; *Brown and Pomeroy*, 1989; *Kinar and Pomeroy*, 2015). Snow traps and particle counters can neither measure the mechanisms of transport initiation nor provide continuous vertical profiles of particle concentration or transport. Yet, most current blowing snow model development has been informed from time-averaged measurements from such devices. Accordingly, simplified models of blowing snow persist in the literature that do not contain self-consistent wind-snow momentum balances, as demonstrated by *Andreotti* (2004) for sand. As well, there is a current lack of detailed measurements of particle-surface interactions in natural conditions.

Recent progress in blowing snow research has been accelerated by novel applications of high-speed imaging systems. *Kobayashi* (1972) pioneered blowing snow recordings with outdoor, 1/8-second shutter speed images. This was the first visual evidence of particle mechanics in the snow saltation layer and was extremely informative in the development of saltation theory (*Pomeroy and Gray*, 1995), but the photographs consisted of blurred snow particle streaks or were saturated with particles, disguising individual particle motions. More recently, *Gordon and Taylor* (2009) designed a novel and effective halogen backlit camera system to effectively obtain particle size and shape parameters in the Arctic, but were limited to an imaging area on the order of 9 mm$^2$. In a further study, *Gordon et al.* (2009) modified this technique to image an area of 124 mm x 101 mm with a black and white binarization algorithm to obtain continuous particle density profiles. Unfortunately, particle velocity measurements were unavailable from either study.

In laboratories, several wind tunnel studies have examined drifting snow with Particle Image Velocimetry (PIV) (*Lu et al.,* 2012; *Tominaga et al.*, 2012), shadowgraphy (*Gromke et al.*, 2014) and shadowgraphic Particle Tracking Velocimetry (PTV) (*Groot Zwaaftink et al.*, 2014; *Paterna et al.*, 2016), providing valuable insights into saltating snow velocity distributions, average relative wind and saltating snow velocities, particle size distributions, qualitative comparisons to Large Eddy Simulation driven transport, and equilibrium wind-blowing snow decoupling. Blowing snow transport model development continues to address small-scale variability (e.g. *Nemoto and Nishimura*, 2004; *Groot Zwaaftink et al.*, 2014), and requires advanced measurement techniques to understand the physics

driving such multi-scale heterogeneities as well as evaluate the uncertainties and assumptions inherent in proposed models.

Of the multitude of blowing snow models that have been developed, many implement components of earlier aeolian saltation or initiation models, e.g. the work of *Bagnold* (1941), *Owen* (1964), *Schmidt*

(1980), *Pomeroy and Gray* (1990), and *Nishimura and Hunt* (2000). In what follows, effort has been made to refer only to the original work containing the model component or measurement campaign under discussion, but comments generally apply to all derivatives. Following the work of *Bagnold* (1941), current theory often represents blowing snow in two layers, saltation and suspension, with a neglected and poorly understood creep mechanism at the lower boundary of saltation (*Pomeroy and*

*Gray*, 1990; *Nishimura and Hunt*, 2000; *Doorschot and Lehning*, 2002). Once the wind surpasses a transport threshold velocity, saltating particles follow ballistic trajectories, and rebound off the surface, rising no higher than 10 cm. As wind speeds increase, saltating particles become suspended by turbulence and disperse upwards. Closely following wind streamlines, suspended particles rarely encounter the ground (*Pomeroy and Male*, 1992; *Bintanja*, 2000).

The two most commonly modeled modes of saltation initiation are *aerodynamic lift*, the direct drag induced ejections of grains, and *splash*, the ejection of grains by rebounding saltating particles (*Doorschot and Lehning*, 2002, *McElwaine et al.,* 2004). However, there are substantial disagreements about these mechanisms; *Schmidt* (1986) calculated that direct aerodynamic lift was not possible under average flow conditions over a level snow surface due to strong snow particle bonding. *Doorschot et al.*

(2004) argued the fragile dendritic snow in their study resulted in aerodynamic lift dominance. It is likely that both mechanisms are possible and that the prevalent mechanism depends on the wind conditions and snow surface structure and cohesion. There is a growing pool of blowing snow models parameterizing these two initiation mechanisms, including the work of *Doorschot and Lehning* (2002), *Nemoto and Nishimura* (2004) and *Groot Zwaaftink et al.* (2014), all adapting the blowing sand

initiation model of *Anderson and Haff* (1991) through wind tunnel measurements.

In contrast to representing saltation as a layer of particles moving with uniform trajectories (e.g. *Owen*, 1964) as is common in snow saltation studies (*Pomeroy and Gray*, 1990; *Tabler*, 1991; *Doorschot and Lehning*, 2002), recent wind tunnel studies and numerical simulations of wind transport of sand have

shown the benefit of representing saltation with continuous grain velocity distribution functions (*Creysells et al.,* 2009; *Ho et al.,* 2012, 2014). From these observations, two populations of saltating particles are distinguishable by kinetic energy rather than by physical properties such as grain size. High-energy particles have higher and longer trajectories that are influenced by changes in wind

strength. However, these particles only constitute the long tails of velocity distribution functions (*Ho et al.,* 2012). The bulk of sand saltation observed in these studies consists of low-energy splashed 'ejecta' and tractating (bed transport) grains undergoing very short hops. These grains generate the majority of mass flux and govern the mean properties of equilibrium saltation (*Ho et al.,* 2014).

It remains unknown how saltation develops, transport mechanics evolve. For instance, in sand, saltation and creep transport

modes are often coupled when saltation begins (*Willets et al.,* 1991): as low-energy surface particles accelerate, they begin feeding upper regions of saltation. Allowing variability of motion in blowing snow saltation models permits consideration of additional mechanisms of saltation initiation and momentum transfer to the snow surface.

It remains unknown how well recent advances in conceptualization of blowing sand transport can

improve descriptions of blowing snow because detailed observations of outdoor blowing snow particle transport processes near the snow surface have not been conducted. Perhaps due to this, current theories of snow saltation are inconsistent with each other and conceptualize a limited range of snow motions and initiation mechanisms. To improve the physical theory of blowing snow initiation and transport, this study demonstrates PTV as a tool for measuring short timescale blowing snow surface motions in

an outdoor environment. The objectives of this study are to examine the mechanics of snow particle motion initiation, the detailed interactions between wind speed fluctuations and snow particle dynamics, and the role of turbulent burst mechanisms that are common in mountain environments in generating shear stress to modify snow saltation. In doing so, the potential for adapting a continuum sand transport model for describing snow saltation particle motions is assessed.

**2 Methods**

Fieldwork was conducted during blowing snow events in March 2015 and February-March 2016 at the Fortress Mountain Snow Laboratory (FMSL), Kananaskis Valley, Alberta, Canada. FMSL receives at

least 800 mm water equivalent of snowfall each winter, can sustain wind speeds exceeding 35 m s$^{-1}$ and is home to several well-instrumented high-altitude, wind-swept observation sites. The blowing snow site (2000 m.a.s.l.) was located in an open base area of the Fortress Mountain ski area (Fig. 1). The area was lightly used, allowing for a 350 m upwind fetch of undisturbed open snowfield, with the foot of a

moderate ridge flanking the west 200 m away. The ground was snow-covered and shrub vegetation buried for the duration of the experiment with snow depths fluctuating from 60 to 120 cm. Two Campbell Scientific CSAT3 three-dimensional ultrasonic anemometers positioned at varying heights depending on snow depth on a single mast (10-40 cm and 140-200 cm) measured wind speed at 50 Hz in three axes.

A unique aspect of this experiment was the implementation of laser-illuminated high-speed videography for outdoor nighttime snow particle tracking observations. A portable rigid frame equipped with a Megaspeed MS85K high-speed camera and a 445 nm wavelength 1.5 W continuous-wave laser was situated on the snow surface less than 1 m downwind from the anemometer mast (March 23, 2015) or 33 cm away perpendicular to the flow (February 3, March 3, 2016). *Dennis and Nickels* (2008) suggest

reasonable application of Taylor's hypothesis up to downstream distances of up to six times the boundary layer depth $\delta$. While there are no measurements of $\delta$ for the present data set, it is safe to assume an extreme value of 1 m is less than $6\delta$ which is often $\mathcal{O}(10-100\,m)$ in the Atmospheric Surface Layer (ASL). Thus, using Taylor's frozen turbulence hypothesis and mean wind speed of the two anemometers as a surrogate for convection velocity, the effect of the downwind separation on the

representativeness of anemometer mast turbulence statistics for the actual location of snow transport is assumed negligible with lag times < 0.25 s. Similarly, for the crosswind orientation, the size of energy containing eddies (discussed in Section 3), even over short recordings, are large compared to the separation. Lags between instantaneous wind and particle velocities are mentioned in the Section 4.

The frame was positioned on the snow surface allowing the camera a perpendicular 30 x 140 mm view

of the flow of saltating snow. Laser light was projected through a cylindrical lens to create a 2 mm wide plane orthogonal to both the snow surface and the view of the camera (Fig. 1). The light plane illuminated a 2D projection of saltating snow particles. This allowed recordings in the lowest 5 cm of the atmosphere, with minimal foreground shadowing and no background reflection. Particle Tracking

Velocimetry (PTV) measurements were calculated by DaVis 8 (LaVision) software and estimated individual snow particle velocities using tracking algorithms that match discrete particles in subsequent frames imaged by the high-speed camera. Particle velocimetry techniques are normally used for wind tunnel studies (e.g. *Zhang et al.,* 2007; *Creyssels et al.,* 2009; *Ho et al.,* 2011, 2012; *Lu et al.,* 2012; *Tominaga et al.,* 2012; *Groot-Zwaaftink et al.,* 2014; *Paterna et al.,* 2016), with few applications, in any discipline, in an outdoor setting (e.g. *Morris et al.,* 2007; *Zhu et al.,* 2007; *Rosi et al.,* 2014; *Toloui et al.,* 2014). This is the first known application of PTV for boundary-layer blowing snow studies in a natural environment.

The high-speed saltation recordings provided a great degree of visual distinction of surface particle motion and the use of 2D laser illumination minimized particle overlap (e.g. *Kobayashi,* 1972). As the camera was focused close to the snow surface, hundreds of thousands of rebound and splash events were recorded over the winter field seasons. In addition to PTV, videos were later reviewed with playback reduced 40-70 times, providing qualitative insight to the mechanics of near-surface saltating particle motion and bed interactions.

Figure 2 displays an example of velocity vectors calculated from 1 second of 23 March 2015 (recording #3). The stationary snow surface was masked out. The dashed black line indicates the height ($h_0$) of the upper limit of particles whose velocities were heavily influenced by surface microtopography and contributed uncharacteristic velocity and concentration profile statistics. In order to account for gradual changes in surface topography, an orthogonal terrain following coordinate system such that $y = 0$ is always at the snow surface and the y-direction is parallel to gravity was adopted to calculate vertical profiles of mean projected horizontal particle velocity $u_p$, and particle number flux concentrations $F_z$

$$F_z = \frac{n(z) * u_p(z)}{\Sigma_z\left(n(z) * u_p(z)\right)}, \tag{1}$$

where $n(z)$ is the number of particles identified at height $z$. This allowed a consistent reference frame along subtle inclines like that found in Fig. 2. Immersed boundary coordinates based on the camera frame $(x_f, y_f)$ are not representative of height above the complex surface, (e.g. $(x_f, y_f) = (5, 100)$ is below the surface whereas $(x_f, y_f) = (1, 5)$ is above the snow) and caused statistical values to become increasingly dubious as one approaches the roughness layer. This can result in misrepresenting surface

fluxes. For example, immersed boundary coordinates indicate a flux maximum of 7 mm for Figure 2 because of the lack of vectors present below this height on the right side of the frame. Additionally, the height of surface microtopography varied as recordings were made over hours of active erosion and deposition, changing the surface structure, and subsequently the relative height of measurements with an immersed coordinate system. Terrain-following coordinates allowed observations to be made over a natural snowpack, crucial for improving the realism of blowing snow measurements, while still accurately defining the near-surface region.

The improved realism afforded by PTV over a natural snowpack in the ASL was counterbalanced by increased difficulty in obtaining valuable data from this methodology and from sonic anemometry during blowing snow storms. Ultrasonic wind speed measurements sometimes included spikes, "NaN" readings or were flagged for skewness/kurtosis (*Vickers and Marht*, 1997); these concurrent video recordings were used only for qualitative comparison. Spanwise fluctuations in wind caused snow particles to travel transverse to the plane of light, and the streamwise wind direction usually varied at the blowing snow site over the course of an evening's observations. PTV relies on particles to remain in the plane of light for illumination and tracking through multiple frames. While the frame could be adjusted for slow variations in wind direction, directional variations during wind gusts were a significant complication. To reduce particle mismatch errors and improve velocity calculation accuracy, initial visual quality controls were implemented, discarding video that contained particles obviously moving transverse to the plane of light.

Post-processing required individual particles to be evident in at least five subsequent frames and limitations were imposed on velocity vector tracks to discard physically unrealistic acceleration or direction change from one frame to the next. The camera depth of field and light plane thickness limited out-of-plane particle velocity components to $\pm 0.5 ms^{-1}$. Further uncertainty derives from the limited ability of PTV software to match individual snow particles at high wind speeds (> 9 m s$^{-1}$ at 40 cm height). The particle matching interrogation area becomes larger as wind speeds increase and particles travel further from one frame to the next. This exponentially increases the number of particles that may be incorrectly matched.

To verify particle enumeration, a dual-threshold black and white binarization technique adapted from *Otsu* (1979) was used to estimate particle concentration in each frame. This complimentary method of particle identification used algorithms that, unlike PTV, are not affected by transverse particle motion or particle matching limitations in gusty conditions. Binarization estimates of blowing snow concentration

profiles were in sufficient agreement with concentration profiles generated by PTV, lending confidence to the measurements of particle trajectories. Additionally, with the binary image and PTV time series, it was possible to use a flood-fill algorithm to identify the connected components of blowing snow particles. Making an assumption of grain sphericity and constant density ($917\ kg\ m^{-3}$) and using instantaneous mean particle velocities, the equivalent diameters of the particles were used to estimate

blowing snow volume fractions and instantaneous density flux $Q_s$ ($kg\ m^{-2}s^{-1}$). Particle diameter measurements generated gamma-distributions of particle size (Fig. 3) consistent with other blowing snow literature (*Budd,* 1966; *Schmidt*, 1982).

PTV measurements in exceptionally high wind speeds (> 10 m s$^{-1}$) were not possible because the laser light became blocked by particles. Therefore the dataset used in this analysis is focused on observations

taken during relatively low mean wind speeds for blowing snow (mean $4 - 7\ m\ s^{-1}$, Table 2); these sometimes included periods of intermittent turbulent bursts and intermittent snow transport. After all post-processing, three nights of recording satisfied all quality controls requirements. This included twelve recordings spanning 266 seconds of raw video and 470,000 frames.

## 3 Results

Examination of data calculated from 23 March, 2015, 3 February and 3 March, 2016 demonstrated the value of PTV measurements over varying wind speeds during periods of natural variation in saltating grain shape, type, and size. Descriptions of the snowpacks following the designations of the International Classification for Seasonal Snow on the Ground (*Fierz et al.,* 2009) for each night can be found in Table 1, with particle size gamma distributions for each recording in Fig. 3. Sample videos

from each night can also be found in the document supplement. During all three nights, transport was highly intermittent, implying wind speeds were near threshold conditions. This was a necessary

condition for accurate particle tracking in aeolian systems as images can become easily saturated (*Ho et al.*, 2014).

## 3.1 Wind Characteristics

Near-neutral (slightly stable) stability conditions were found during all nights using flux and gradient based methods (document supplement Fig. 1-3), however, steady-state wind conditions $\left(\frac{\partial U}{\partial t} = 0\right)$ did not occur during the field campaign. The less strict steady-state requirements of *Foken and Wichura* (1996) were also tested to further confirm steady-state conditions were not evident. Recording and wind characteristics encompassing the three nights are displayed in Table 2. Fifteen minute and recording time period mean wind speed $\bar{u}$, friction velocity $u_*^2 = \left[\overline{u'w'}^2 + \overline{v'w'}^2\right]^{1/2}$ (*Stull*, 1988), and covariance based roughness length $z_0 = z\, e^{-0.4\bar{u}/u_*}$ are shown for both anemometer measurements as they are the parameters most often used in blowing snow models. Additional values of turbulence intensity $I = \sqrt{\overline{u'^2} + \overline{v'^2} + \overline{w'^2}}/\sqrt{\overline{u^2} + \overline{v^2} + \overline{w^2}}$; and Shields number $S = \rho_{air} u_*^2 / \rho_{ice}\, gd$ (based on mean particle size for each video) over both time periods are provided, as well as mean blowing snow flux ($Q_s\ kg\ m^{-2}s^{-1}$) for the recordings.

If wind measurements are close to the surface, such as during the 3 March 2016 recordings, the physical path length of the sonic anemometers can result in losses of high frequency turbulence. Following the guidelines of *van Boxel et al.,* (2004), the Nyquist frequency (25 Hz) is a limiting factor for mean wind speeds greater than $3\ m\ s^{-1}$, and may also contribute to some discrepancy of low and upper anemometer turbulence measurements. Additionally, the lower anemometer measurements during Recording #2 on March 23, 2015 appears to have been contaminated as there is a significant change in covariance derived $u_*$ and $z_0$ values between the two heights.

The ASL fit a Prandtl-von Kármán logarithmic-law profile poorly during the storms, most likely due to violations of horizontal-homogeneous-flat (HHF) terrain and stationarity requirements. Recording period log-law based roughness lengths $z_0 = e^{\frac{u_1 \ln(z_2) - u_2 \ln(z_1)}{u_1 - u_2}}$ and friction velocities $u_* = \frac{\kappa \bar{u}(z)}{\ln(z/z_0)}$ were

loosely comparable to lower anemometer fluctuation-based measurements with $z_0$ errors less than $\pm\ 100\%$ (except March 3 #9), and $u_*$ errors less than $\pm\ 70\%$, often slightly underestimating.

The 15-minute roughness lengths that were generated by covariance methods resulted in inaccurate log-linear wind profiles that indicated a zero velocity zone for the wind well above the snow saltation layer and often at extremes values of tens to hundreds of mm (Table 2). High roughness lengths appear characteristic of this mountain region. At a nearby site 14 km northeast and 600 m lower in elevation, *Helgason and Pomeroy* (2005) attributed similar large covariance derived $z_0$ values at varying heights to the effects of surrounding topography and the non-stationary and non-steady state nature of the wind. The modified "focal-point" log-linear wind profile proposed by *Bagnold* (1941) for aeolian transport was not recognized in this study, with estimates of focal lengths fluctuating from several mm up to 6 m. The wind was characterized by brief moments of intense gusting separated by periods of relatively calm conditions as also noted at the *Helgason and Pomeroy* (2005) research site. 15-minute turbulence intensity ranged from 26 to 113%, consistently higher than the recording period values where short time series preclude larger fluctuations around mean values. As a result, values of $\bar{u}$, $u_*$, and $z_0$ consistently differ between video recording-averaged (7.3 to 28 s) and 15-minute averaged values. Turbulent gust-driven snow transport events dominated the nights. 11 out of 12 fifteen-minute averages present lower wind speeds than the recording period with the long averages often below thresholds of transport.

Figure 4 shows varying Reynolds Stress ($RS = u'w'$) generation during recording specific periods following the language of quadrant hole analysis (*Willmarth and Lu*, 1971). Sweep and ejection events (Q2 and Q4) often contributed the majority of RS at both anemometers, with a more pronounced role closer to the ground, indicating changes in the snow surface influence on wind mechanics. Q2 and Q4 stress also occupied a disproportionately small amount of time near the surface, as can be seen in the impact factors inset in the bar graphs ($IF = (\%\ Reynolds\ Stress)/(\%\ Time)$) that are greater than unity. Therefore, when strong events are captured during the recordings, RS values can be much larger than long time averages. The presence of a single pronounced sweep event in Recording #3, March 23 (discussed in Section 3.3) contributed to a high recording period turbulence intensity (40-45%), and a much higher recording friction velocity than the 15-minute values (0.48 m/s and 0.24 m/s respectively) and will be discussed in detail in Section 3.3.

Understanding the changes in the quadrants generating RS helps illuminate the differences in $u_*$ values at the two measurement heights over these short recording timescales. The recordings with the largest discrepancy in $u_*$ (besides March 23 #2 where low height wind measurements are questionable) are February 3 #3 ($u_* = \{0.30, 0.45\} \, m \, s^{-1}$) and March 3 #2 ($u_* = \{0.26, 0.43\} \, m \, s^{-1}$). This was the

result of a significant decrease in the magnitude of mean RS at the lower measurements (0.27 vs 0.11 $m^2 \, s^{-2}$ and 0.22 vs 0.09 $m^2 \, s^{-2}$, February 3 and March 3, respectively), while the turbulence intensity remained nearly constant (Table 2). The reduced presence of Q1 and Q3 at the lower heights, and increased impact factor of Q2 and Q4 (Fig. 4) indicated a complex shift in boundary layer dynamics towards the snow surface that is beyond the scope of this manuscript. Other recordings exhibited much

closer friction velocity and roughness length values at the two anemometers, indicating similar turbulent motions were captured.

As also seen by *Bauer et al.* (1998), sweeps and ejections did not immediately follow one another, rather there were prolonged clusters of sweeps and ejections with gaps in between (Fig. 5a). The gaps may be a result of point measurements' inability to capture a full 3D motion (*Bauer et al.*, 1998), but

nevertheless the measurements showed significantly different RS generation than that typically found in wind tunnels. For example, Fig. 5 a-b compares RS values in a recent blowing snow wind tunnel experiment of *Paterna et al.* (2016) with RS found on 3 February 2016 at FMSL at similar friction velocities ($0.25 \, m \, s^{-1}$ and $0.27 \, m \, s^{-1}$, respectively). A sweep signal of magnitude greater than one standard deviation of RS is indicated above the given RS time series by a blue triangle, while similar

ejections are marked by brown triangles. Visually, there is a noticeable shift toward clustered sweep and ejection events at FMSL (Fig. 5a), in which clustered pockets of sweeps alternated with ejections, while the sweep/ejection cycle and turbulent energy occurred at much a higher frequency in wind tunnel-based measurements by *Paterna et al.* (2016) (Fig. 5b). This is further confirmed in the power spectral density plots of streamwise wind speed (Fig. 5c), and in the discussion by *Paterna et al.* (2016).

Reconciling these differences between motions in atmospheric boundary layers and wind tunnel flows is challenging (*Hutchins et al.*, 2012) and the substantial differences in Reynolds number must be kept in mind when comparing blowing snow studies in wind tunnel and outdoor environments.

## 3.2 Vertical PTV Profiles

Figure 6 shows profiles of ascending particle horizontal velocity for the three nights of recording with linear regressions based on the lower 10 mm. Profiles were designated by their recording-specific low anemometer $u_*$ values, except for 23 March where the lower wind measurements for recording #2 were contaminated. Thus, all 23 March recordings were compared by 2 m wind.

Particle motions begin with an initial ejection velocity at the surface and then accelerate due to fluid drag in the wind. Therefore the height of an ascending saltating particle should be a function of the time spent accelerating. The profiles of horizontal velocities of ascending particles confirmed this acceleration in Fig. 6. The average momentum transfer from wind to grain was estimated from the inverse slope of the plots and indicated the ability of the wind to entrain and accelerate particles. For all three nights there is a near constant particle velocity gradient immediately above the surface, $\left( \frac{\partial u_p}{\partial z} = \gamma, \gamma \in \mathbb{R}^+ \right)$. Above ~8-12 mm, depending on the night, mean particle velocities deviate from the linear profile as seen in Fig. 6 and confirmed by normalized root mean square error (NRMSE) changes from the order of 0.01 to 0.1 above and below 10 mm, respectively. It must be noted, this did not indicate a discrete transition height from creep to saltation but is rather evidence of a continuous spectrum of particle velocities (*Anderson*, 1987; *Ho et al.*, 2012) transitioning to a higher energy population away from the surface. As these recordings captured intermittent transport, saltation is in constant readjustment to the turbulent wind, with particles falling in and out of the higher levels of saltation (discussed further in Section 3.3). This prevented a consistent adherence to the linear profile as seen by *Ho et al.* (2011), though both studies found linear profiles overestimate particle velocity at greater heights.

The velocity gradient (shear rate) $\gamma$, was estimated by linear regression and varied from $35 - 98 \ s^{-1}$. Variations in wind speed and $Q_s$ during each recording period indicate blowing snow transport never attained equilibrium. However, $\gamma$ values are comparable to wind tunnel PTV sand velocity gradients ($39.0 - 150 \ s^{-1}$) measured below 30 mm by *Zhang et al.* (2007), and the range $20 - 60 \ s^{-1}$ found by *Ho et al.* (2011). For each night, $\gamma$ increases with increasing friction velocity (Fig. 6d-f). The *Ho et al.* (2011) rigid bed experiments were conducted at comparable Shields numbers to the high HHI, wind-

hardened February 3 experiments (Ho et al.: [0.013, 0.043], Aksamit and Pomeroy: [0.026, 0.061]) and shared several trends discussed here and below. For example, as for *Ho et al.* (2011) the night of February 3 had on average the lowest $\gamma$ values (mean $44\ s^{-1}$ versus $69\ s^{-1}$ for the erodible beds) and the least variation in $\gamma$ though transport occurred during comparable friction velocities, and higher

Shields parameters than many March 23 and March 3 recordings (Table 2). The erodible bed studies were performed at consistently higher Shields parameters for the Ho experiments (Ho et al.: [0.07, 0.14], Aksamit and Pomeroy: [0.01, 0.12]), yet both studies also found increases in $\gamma$ with friction velocity for the erodible beds. *Ho et al.* (2011) found less variance in $\gamma$ over all friction velocities as could be expected from consistent equilibrium conditions.

As with wind tunnel sand studies (*Zhang et al.*, 2007; *Creyssels et al.*, 2009; *Ho et al.,* 2011), large non-zero particle slip velocities were observed. The influence of surface microtopography and density of the flow prevented an exact measurement of particle slip velocity $u_0$, because it becomes difficult to enumerate all grains at the surface (*Creyssels et al.* 2009). However, extrapolating the linear regression plots of constant shear rate $du_p/dz = \gamma$ one can estimate $u_0$. As found in the same Ho study, our

February 3 "rigid bed" experiments exhibited a nearly linear increase in $u_0$ with $u_*$ (Fig. 6h). While our $u_0$ measurements had a larger range for the "erodible bed" nights of March 23 and March 3 (Fig. 6g, 6i) than that of *Ho et al.* (2011) and *Creyssels et al.* (2009) who found a near constant slip velocity, no definitive trend with $u_*$ could be identified either. A purely constant slip velocity over erodible beds most likely depends on equilibrium transport conditions as has been theoretically explained by *Ungar*

*and Haff* (1987) and may explain the ambiguity in these results.

Figure 7 shows the vertical profile of the normalized particle number flux calculated as Eq. (1). A normalized number flux profile was chosen instead of the volume fraction (e.g. *Ho et al.,* 2011) or mass flux density profile (*Creyssels et al.,* 2009) because of computational limitations of the PTV package in DaVis 8, and because of the non-equilibrium transport during the recordings. Since it is impossible to

control the rate of transport in nature, and volume fractions will change with rates of transport, wind fluctuations and snow surface conditions, it was informative to compare number flux concentrations between periods of diverse mass transport to determine differing transport mechanics over varying snow and wind conditions. As the study was focused on the dynamic role of surface transport, and not

measuring mass flux, each concentration profile was renormalized by the amount of flux that occurred during a recording to compare what proportion of total particle transport is occurring at each height at suggested by *Ellis et al.* (2009) for aeolian transport profiles in nature. This allowed observation of changes in the relative importance of regions of transport.

The fractional number flux fits an exponential decrease of the form $v(z) = v_0 \exp(-z/l_v)$ with increasing accuracy as one approaches the densest flow at the surface, similar to sand and snow saltation profiles seen elsewhere (e.g. *Maeno et al.*, 1980; *Nishimura and Hunt*, 2000; *Creyssels et al.*, 2009; *Ellis et al.,* 2009; *Ho et al.*, 2011; *Lü et al.,* 2012), with $v_0$ and $l_v$ being fitted parameters, the latter referred to as the decay length. The number flux decay length $l_v$ indicated how quickly the

number flux concentration approached zero (Fig. 7d-f), but because there were large variations in the surface concentration $v_0$ in the present study, more consistent trends can be observed with the momentum deficit height $h_v$, the height below which 75% of particle flux occured. As seen in the right inset of Fig. 7, there is a non-linear increasing relationship between friction velocity and $h_v$ for the February 3 and March 3 recordings. After disregarding March 23, (discussed below), values of $h_v$

followed an approximate power law relationship ($au_*^b + c, R^2 = 0.77$) with asymptotic-like behaviour towards the top of the region of interest (~11 mm). At low friction velocities, near-surface saltation was dominated by transport below 7 mm, with transport becoming gradually more uniform as $h_v$ approached the top of the camera frame at higher friction velocities.

March 23 exhibited very little change in $h_v$ and $v_0$ with only a slight decrease in $l_v$. Thus, concentration

was largely invariant with wind strength. This is remarkably similar to the erodible bed findings of *Ho et al.* (2011). Only one recording had comparable Shields numbers to Ho, but the non-cohesive graupel bed, and spherical snow grains most similarly represented sand grains and an erodible sand bed out of the three nights.

The near surface location of the maximum of $F_z$ found over all recordings in Fig. 7 is in disagreement

with models using Bagnold's focal height (*Bagnold*, 1941) to predict a peak mass flux at some distinct height above the surface (e.g. *Pomeroy and Gray,* 1990). This stemmed from the earlier lack of high resolution measurements of near snow surface processes outdoors, as results were in agreement with later wind tunnel observations (*Sugiura et al.*, 1998; *Nishimura and Hunt*, 2000) and numerical studies

(*Nemoto and Nishimura,* 2004) of snow flux profiles as well as the recently measured blowing snow density profiles *Gordon and Taylor* (2009) and *Gordon et al.* (2009) found over natural snowcovers in Churchill and Franklin Bay, Canada, respectively.

For the time series investigated, mean particle diameters had small temporal variance over any given recording (0.01 mm on Mar 23, 0.05 mm on Feb 3, and 0.02 mm on Mar 3). For a given time step, mean particle diameters tended to decrease with height in the field of view, typical of saltating snow studies (e.g. *Sugiura et al.*, 1998; *Gromke et al.*, 2014), with the most extreme variations on the order of 60 *μm*. From this it can be inferred that particle number concentrations were closely related to particle volume fractions through mean diameters, and one can neglect variations in particle size with wind speed changes while admittedly underestimating the relative volume concentration close to the surface.

Particles moving in the densest region of the flow, immediately above the surface, are in a zone where particle tracking by opto-electronic snow particle counters becomes impossible but PTV provides new information. Close to the surface, it is possible to observe the whole spectrum of saltating particle velocities including those considered to be moving via creep.

Similar to the high and low-energy saltating grain populations theory of *Ho et al.*, (2014), terrain following height bands were chosen such that two end-case populations could be delimited. Unique descending particles were separated into upper and lower regions according to a variable boundary (2 to 5 mm) so that they appear at most once in each region. Then, particles were binned into 20 equivalent size horizontal velocity bins. Figure 8 shows one example of histograms generated by these bins for a given height separation (3 mm for Mar 23 and Mar 3, 4 mm for February 3). Assuming that most descending particles tracked in the lowest 30 mm (frame size) will impact the surface, then most descending grains in the upper region will also be present in the lower region. This behaviour appears in Figure 8 where the upper region population is a subset of the lower region population, showing high and low-energy populations coexist as part of a continuous spectrum of motion at the surface. This is indicative of an inherent coupling of the creep and saltating grain motions. There is sensitivity to the selection of upper and lower regions. With a separation threshold below 2 mm, numbers of high velocity low-region particles are underestimated because tracking is increasingly difficult. With the separation threshold above 5 mm, the low density of particles and sensitivity to out of plane wind

fluctuations made measurements of representative fluxes inaccurate. Thus, the separation threshold was restricted to the 2-5 mm range.

   For every recording and each separation threshold chosen, there was a denser surface flow whose mean statistics are dominated by slow moving particles. This is to be expected from the particle velocity and number flux profiles in Fig. 6 and 7. The upper region histograms show saltating particles starting to transition towards higher energy trajectories, with transport dynamics dominated by larger horizontal velocities. The increasing proportion of high-energy particles with distance from the surface is likely due to the need for a greater velocity to reach greater heights on a ballistic trajectory from the surface and the subjection to higher wind speeds with increasing distance above the surface.

Subtracting the upper region particle bin numbers from the lower region bins permits an estimate of the number of particles at given velocities present solely in the lower region, hereon termed the surface residual. The sensitivity analysis showed that for nearly every separation threshold at least as many high velocity particles existed in the upper region as in the lower. This allows conceptualization of the surface residual as the slower moving surface grains, though not necessarily creep. An example of this is shown in histograms of descending lower-region, upper-region, and surface residual particle horizontal velocities for March 3 recording #5 in Figure 8.

Assuming a fixed particle diameter for all particles in each recording as the mean from Figure 3, spherical snow grains and a grain density of $917 \ kg \ m^{-3}$, snow transport momentum for each region could be estimated as the sum of momentum of particles in each velocity bin. Varying upper and lower region separation thresholds from 2 to 5 mm, the surface residual constituted between $2 - 82 \ \%$, $0.07 - 35 \ \%$, and $5 - 49 \ \%$ of total transport momentum below 30 mm, on March 23, Feb 3, and March 3, respectively. Ranges of momentum contribution for individual recordings are indicated on the respective histograms in Figure 8 as "Surf Res," and are plotted in the bottom left panel. Surface residual momentum values compliment the profiles of $F_z$ (Fig. 7) as near surface $F_z$ values also include high velocity particles that impact the surface.

In the near-surface region, the ability of the snowpack to redistribute impact momentum was estimated from PTV data derived from each recording. Mean snow particle rebound efficiency varied from night to night, and was quantified by the restitution coefficient, $\overline{e_{xz}} = \overline{\|s_r\|}/\overline{\|s_\iota\|}$, where $\overline{\|s_r\|}$ and $\overline{\|s_\iota\|}$ are

mean ejection and impact speeds of particles, respectively, at 6 mm $\pm$ 2 mm such that the lower bound of the measurement band corresponds with the upper bound of the surface band generating Fig. 8 histograms. Because of the density of the particle flow, and transverse components of travel, a bulk statistical approach must be used to quantify momentum redistribution into the particle bed. Therefore

particle ejection speeds of both the rebounding grains and the splashed grains that reach ~20 particle diameters above the surface were averaged. Over the course of the campaign, $\overline{e_{xz}}$ varied from 0.58 to 0.84, within the bounds of the previous wind tunnel blowing snow study of *Sugiura and Maeno* (2000) who used a complimentary particle by particle approach, separating individual rebounding and splashed grains. The mean restitution coefficient was 0.69 for the graupel grains on March 23, 0.79 for fresh

snow on March 3, and 0.73 for old snow on February 3. This suggests rebound efficiency was dependent on time-sensitive saltating snow crystal and bed mechanical/material properties, as also noted in a blowing snow wind tunnel study by *McElwaine et al.* (2004).

### 3.3 Turbulent Event Transport

The initiation mechanisms observed at the surface during the onset of transport events differ from those

proposed in single threshold velocity models (e.g. *Schmidt*, 1980), suggesting multiple thresholds with the dense surface flow playing a crucial role. All three transport thresholds recognized during video playback were crossed during 23 March recording #3 (Fig. 9) where an isolated gust was captured with minimal antecedent transport. Thus, it will be used as an example for further discussion. Concurrent streamwise wind measurements at 200 and 40 cm are plotted in Fig. 9a showing penetration of a

turbulent sweep to the surface that is responsible for snow transport. In Fig. 9a, filled circles indicate sweep events with RS exceeding one standard deviation of total RS (colors corresponding to measurement heights), while triangles indicate similar moments of strong ejections. Fig. 9b and 9c show time series of spatially averaged particle velocities, and total particles tracked, respectively, in three height bands, $1 < z < 4\ mm$ (Near-Surface), $4 \leq z < 8\ mm$ (Mid), and $8 \leq z < 30\ mm$ (High).

These three heights were chosen to demonstrate the subtle differences in particle transport and the continuum of motion as grains began motion and began bouncing to greater heights as wind speeds increased. These are not hard thresholds of "creep" versus "saltation" regimes. Fig. 9d shows the time

series of instantaneous blowing snow flux $Q_s$ in $kg \, m^{-2} s^{-1}$. These binarization based flux measurements compliment PTV calculations in intense gusting when enumerating all particles through tracking became difficult.

At the end of a strong ejection event (2.5-4.5 s) at wind speeds near $4 \, m \, s^{-1}$, snow particle motion began with tumbling surface movement where aerodynamic drag was barely able to directly break weak surface crystal bonds and initiate rolling (5 s in Figs. 9b-d). Particle-bed collisions were concurrently responsible for breaking surface snowpack matrix structures at these wind speeds, though were not yet able to initiate a splash regime. The supplementary video for March 23 begins at 5 seconds in this time series. The bonds broken by low-energy grains at low wind speeds enabled more grains to be freely available for entrainment. During this time, horizontal particle velocities remained low and in the near surface region (Fig. 9b), with no particles being tracked above 4 mm (Fig. 9c), and total mass transport remaining low (Fig. 9d). At this stage, the only particles in motion were those classically termed creep.

As the wind speed increased (> 6 s), another threshold was crossed ($\sim 4.5 - 5 \, m \, s^{-1}$) above which tumbling near-surface particles were sufficiently accelerated so that they could regularly bounce off the uneven surface and out of the creep layer. This initiated what is classically described as saltation, evidenced by the increasing presence of particles tracked in the "Mid" and "High" regions in Fig. 9c (above 4 mm), though snow mass flux remained moderate at this time (Fig. 9d). A continuum of motion is evidenced here as all mean particle velocities increased and velocities steadily increased away from the surface.

At 8 s, a strong sweep is present at 2 m that penetrates to the surface by 8.5 s when the last critical wind velocity threshold was crossed ($\sim 6 \, m \, s^{-1}$); that at which saltating particles were sufficiently accelerated to initiate an active splash regime upon rebounding. At this point snow mass flux increased exponentially (Fig. 9d), abruptly saturating the recording frame with snow particles and limiting illumination for successful PTV (discussed below). Similar exponential increases of sand flux during gust onset and splash commencement have been documented (*Willetts et al.*, 1991). The increased snow mass flux (Fig. 9d) persisted for the duration of the gust, (until 10 s) after which both high and low streamwise velocities decreased. From 11 s onwards, the decreasing wind speed was no longer able to sustain the mass transport and particle counts dropped in the "Mid" and "High" regions of flow. A

combination of inertia and wind drag prolonged transport in the creep layer, maintaining rolling crystals that continued breaking surface bonds and were available for transport during the next gust.

High region particle velocity spikes occurred with some delay after 8.5 s due to the intense snow particle density blocked the laser light illumination, making particle tracking difficult. At 9 s, the

number of particles tracked in the upper region increased as particles tracked near the surface decreased. This is likely a measurement error due to the dense granular flow attenuating light penetration to the snow surface. As the gust began to subside after 9 s, PTV-observed velocities and particle numbers increased because tracking became more successful. Then, as wind speed decreased further, observed velocities and particle numbers decreased as expected. The *Otsu* (1979) binarization thresholds were

determined over short sub-periods (0.085 s) of recording #3 (Fig. 9d), allowing the thresholding technique to adapt to different levels of illumination and overcome these saturation issues.

Particle impact dynamics evolved as snowpack surface conditions varied during the season with multiple melt-freeze cycles, periods of wind hardening and the appearance of mixed grain types. Warm (Air Temp: +1°C) February 2015 snowstorms precipitated enormous aggregate and rimed crystals that

expanded the role of near-surface particle dynamics. Large (4 mm diameter) tumbleweed-like aggregate grains termed here "tumblons", eroded many smaller crystals from the surface or shattered themselves and immediately became saltating grains, depending on impact velocity. Overlain PTV vectors can be seen in Supplementary Video – Tumblon PTV, where an impacting tumblon shatters at the surface at seven seconds, with an impact velocity of approximately $(u, v) = (2.46, -0.43)\ m\ s^{-1}$. At 35 seconds

in the PTV video, a comparably sized tumblon with velocity $(u, v) = (0.6, 0.1)$ m s$^{-1}$ tumbles across the screen without collapse. This type of particle motion has not been described before and would seem to be a distinctive feature of blowing snow during or shortly after snowfall of large dendritic flakes in relatively warm conditions. Uniquely large grains can also be found in the March 3 supplementary video, though shattering dynamics do not appear prevalent on that night as large grains resulted from

riming and not wet grain aggregation. Decomposing and aggregate grains of extreme size have not been reported for saltating sand. This may limit the application of sand bed momentum balances and wind tunnel studies where there are no contributions of falling snow to saltation.

**4 Discussion**

Choosing the appropriate timescale to characterize turbulent energy for snow transport is vital. From Table 2 and Fig. 6 & 7, it is clear that recording specific particle velocity gradients $\gamma$, and flux concentration profiles did not scale with 15-minute $u_*$ or $z_0$ values as is assumed on average for many existing snow saltation models (*Pomeroy and Gray*, 1990). Part of this lack of concurrence was due to intermittent transport and large-scale atmospheric motions generating high shear over short periods (Fig. 4). Also, the limited Reynolds numbers possible in wind tunnel blowing snow experiments cannot replicate the complex eddy structure of the ASL. Thus kinetic energy is contributed to mass transport at much higher frequencies in wind tunnels (*Paterna et al.*, 2016) and the full spectrum of motion can be measured over shorter time scales. Moreover, with the influence of surrounding topography, capturing the relevant range of energy containing eddies to predict snow transport in the alpine is less straightforward. The 15-minute mean wind speeds were often below any snow transport thresholds reported in the literature (*Li and Pomeroy*, 1997). The significant errors arising when applying time-averaged values in a $u_*$ driven transport model (i.e. *Bagnold*, 1941) in intermittent winds have been well examined for sand (*Sorenson*, 1997) and equally apply for snow saltation. *Pomeroy and Li* (2000) accounted for the inapplicability of steady-state theory in near threshold conditions by using a probability of occurrence function to reduce transport fluxes at lower mean wind speeds, but it is unclear whether this empirical correction can account for the complex interaction of turbulence and particle flux near the threshold. Disagreements between $u_*$ and $z_0$ values at the two measurement heights, large 15-minute $z_0$ values, and disagreement between log-law based and covariance generated $u_*$ values reinforces the notion that all required assumptions must be met before log-law profiles should be applied for blowing snow models (*George*, 2007), especially in complex terrain.

Blowing snow PTV transport profiles were clearly more related to recording-specific $u_*$ than 15-minute values because of the non-steady state nature of the wind. As found in several wind tunnel sand studies (*Creyssels et al.,* 2009; *Ho et al.*, 2011), particle velocity gradients adhered to linear profiles below ~8-12 mm depending on the night. The low-energy near surface population of particles were less affected by fluctuations in wind strength (*Ho et al.*, 2014), which resulted in more temporally consistent periods of near surface low-energy transport and lower NRMSE values. Recent blowing snow wind tunnel

experiments found log-linearity present in the horizontal velocity profile (*Tominaga et al.*, 2013), though this was not observed in the present study because of a lack of presence of a log-law for the wind.

For all recordings, the velocity gradients $\gamma$ increased with increasing friction velocity estimates (Fig. 6d-f). Thus even in the dense low-energy population of grains, there was noticeable adaptation to changing wind speeds. However the role of low-energy grains diminished as surface grains became less available with increasing surface hardness. As with the rigid bed experiments from *Ho et al.*, (2011) at similar Shields parameters, there was a much smaller variability in $\gamma$ for the wind-hardened bed on February 3 than for the two nights with lower HHI and lower overall $\gamma$ values. The momentum deficit height $h_v$ was also highest for February 3 (Fig. 7 inset) with relatively low Shields numbers (0.026-0.061), indicating a muted role of creep and a more uniform saltation layer.

On the same night, a linear increase in particle slip velocity $u_0$ was observed for wind-hardened beds, similar to the findings over rigid beds by *Ho et al.*, (2011) for sand. This showed a general acceleration of all grains in saltation when there are fewer new grains to be entrained in the flow. In comparison, there was much less variability for slip velocity for March 23 and March 3 over a wider range of Shields parameters and no clear increasing or decreasing trend. A constant slip velocity is characteristic of erodible beds in equilibrium sand transport experiments (*Creyssels et al.*, 2009, *Ho et al.*, 2011). Presumably, as increasing concentration compensates for increasing friction velocity, $u_0$ returns to mean values over time. Snow transport observed to be in constant readjustment to changes in wind speed, and equilibrium was never reached in the experiments, though reduced variation in $u_0$ may be indicative of early stages of the equilibrium process as theorized by *Ungar and Haff* (1987).

Particle number flux concentration profiles fit an exponential decrease model with increasing accuracy as height decreased and flow density increased, as predicted in many other sand and snow studies (*Maeno et al.*, 1980; *Nishimura and Hunt*, 2000; *Creyssels et al.*, 2009; *Ellis et al.*, 2009; *Ho et al.*, 2011; *Lü et al.*, 2012). A direct comparison of decay length ($l_v$) between recordings and other experiments was impractical because large variations in surface concentration ($v_0$) skewed the decay length, making it a poor analogue for saltation height. Instead, it was found that the momentum deficit height $h_v$ increased for most recordings with increasing friction velocity. Lower $h_v$ values require more

particle transport momentum nearer the surface. The presence of low-energy near-surface particles at low friction velocities served as a reservoir for the transition to saltation with subsequent increases in wind speed. As the wind speed accelerates, more particles are accelerated and transported to greater heights where they are further accelerated, resulting in a more uniform vertical distribution of mass flux.

This occurs gradually with increasing wind speed, rather than involving discrete transport threshold velocity values for separate modes of transport.

The notable exception for this trend in $h_v$ were the events of March 23 (Fig. 7, inset – red dots) where $h_v$, $l_v$, and $v_0$ values remained constant as was predicted by *Ho et al.* (2011) for erodible beds. This difference in behaviour can be physically justified as spherical graupel grains over a non-cohesive bed

are the conditions most closely resembling sand, and so the number flux profiles behaved more similarly to sand than for the other two nights of blowing snow.

Histograms of horizontal velocity further supported the relevant and dynamic role of the low-energy surface population of blowing snow (Fig. 8), and the decreasing importance of near-surface mass flux with increasing surface hardness. The bottom left plot in Figure 8 shows that for the nights with

erodible beds and lower HHI (March 23 and March 3), the ratio of surface residual to total momentum is largest for low friction velocity. This is not sensitive to the separation threshold height chosen. During low wind speeds, particle transport for the erodible beds was largely confined to the lower region with few high-energy surface impacts. The ratio of surface residual to total momentum, and therefore number flux, was much higher over erodible beds than over the wind-hardened rigid bed

recordings on February 3 (Fig. 7 & 8). This complimented the fact that the low-energy surface transport on February 3 had the smallest contribution to total number flux. Thus particle type and snow bed properties played a significant role in the surface momentum balance, changing the uniformity of saltation profiles and wind momentum lost to surface transport. Mean particle diameters remained relatively similar over all recordings and thus snow transport models need to account for snow bed

hardness or erodibility as there is a connection to a variable near surface transport momentum sink.

Analyzing the instantaneous wind speeds in Fig. 9a helps to explain the short timescale roles of gusts, high friction velocity and turbulence intensity for snow transport in recording #3. The turbulent sweep event ($u' > 0, w' < 0$) from 8 to 9 s generated considerable RS; this turbulent structure is widely

reported to be involved in initiating aeolian sediment transport (*Grass,* 1971; *Jackson,* 1976; *Sterk et al.* 1998; *Chapman et al.,* 2012).  The 1 s sweep accounted for 29% and 25% of total absolute RS at 40 and 200 cm, respectively, and contributed 56% of total snow particle flux below 30 mm, but occupied only 8% of the time.   Turbulent ejections ($u' < 0, w' > 0$) generated 39% of total absolute shear stress during the recording and contributed the same direction of RS values to friction velocity calculations but only resulted in 3% of total snow particle flux. Varying the lag time between wind and snow measurements from 0 to 1 second to determine the resultant snow flux had no significant effect on these calculations.

In this gusty alpine environment, periodic turbulent gusts generated the majority of momentum flux as seen by impact factors greater than unity at the surface for Q2 and Q4, and small contributions from Q1 and Q3 (Fig. 4). More importantly, recording #3 showed that the sweep event with strong positive $u'$ fluctuation resulted in particle entrainment and transport, whereas the large ejection event was ineffective at generating snow transport. As suggested by *Sterk et al.* (1998), instantaneous wind speed is a potentially better predictor of snow transport than friction velocity. However, during each night, the mass flux for scaled with increasing friction velocity (Table 2). This resiliency may help explain some of the robustness of $u_*$-based time-averaged uniform trajectory blowing snow models, but requires further investigation. The role of sweep events such as this for initiating snow saltation is potentially important for developing models that couple turbulence to snow erosion, entrainment and mass flux and may help resolve current uncertainty in estimating threshold conditions for transport. The importance of understanding snow response to instantaneous wind speed is further increased in complex terrain where the 300 m of clear upwind fetch or 60 s of constant wind suggested by *Takeuchi* (1980) for saltation to fully develop is not always available. For a more general application, this requires further investigation of turbulent snow transport over longer time series and other snow conditions.

Designating creep as distinct from saltation as originally done by *Bagnold* (1941) is not only unnecessary but also physically inaccurate as snow transport displays a continuous spectrum of motions, similar to that proposed for sand by *Anders*on (1987), and individual particles can easily transition from one form of motion to another over their trajectories. However, two populations of motion in this spectrum could be identified in the analysis of Fig. 6-9. While undergoing wind speed

fluctuations, the region of near-surface flux has the most temporally consistent transport (Fig. 9). This was also the region of largest velocity variance (Fig. 8), yet the consistent presence of the slow particle flow allowed the best fit of particle number flux and mean particle velocity to profiles suggested in the equilibrium sand transport literature. The worst fit of both gradients always occurred at the upper

regions dominated by high-energy particles (Fig. 8). These high-energy particles constituted the population of fast moving grains that was most susceptible to changes in wind speed (*Ho et al.* 2014) and most temporally intermittent (Fig. 9).

The lower boundary condition for momentum transfer is complex due to creep and dependent on instantaneous wind speed and turbulent motions near the surface.  As a result, equilibrium conditions

were never found in the field observations reported here. Non-equilibrium saltation-wind interactions cannot be described with simple uniform trajectories. The majority of particle trajectories in saltation consist of short hop lengths and times, resulting in high frequencies of particle collisions that break surface bond structures, and create dense quasi-fluidized bed characteristics. The complexity of conservation of mass, momentum, and kinetic energy in blowing snow in natural environments, such as

measured here, cannot be understated, especially when the large rimed, aggregate tumblons were present. In the alpine snowpacks investigated, variable particle restitution coefficients contributed to this complexity. While high HHI wind-hardened surfaces exhibited similar behaviour of slip velocity and particle velocity gradients as rigid bed sand studies, complexities over a natural snowpack prevented conclusive bimodal "erodible" versus "non-erodible" scale relations that appear to be viable

for sand (*Ho et al.,* 2011).  Blowing snow is a distinctive two-phase flow.

Despite arguments to the contrary for other materials (e.g. *Sterk et al.,* 1998), saltating snow models relate mass flux to surface shear stress estimated from air motions well above the surface. These estimates are often based on a momentum deficit derived from the total number of particles in transport, neglecting the vertical heterogeneity of particle concentration and momentum within the snow saltation

layer (*Doorschot and Lehning*, 2002). As all panels in Fig. 6 and 7 show, uniform descriptions of surface shear stress calculations based on concentration and flux measurements above 10 mm overlook the substantial wind momentum transferred into the creep layer (i.e. *Zhang et al.,* 2007, *Creyssels et al.,* 2009; *Ho et al.,* 2011, 2012, 2014). Disregarding this flux prevents calculation of the full momentum

balance. Accounting for variability in saltation trajectories would also allow for a dense surface flow to be represented that can feed upper regions (e.g. *Nemoto and Nishimura*, 2004) and create a self-consistent momentum balance. *Andreotti* (2004) wrote a further discussion of self-consistency errors of wind feedback in single trajectory saltation models.

The wide variety of snow saltation initiation mechanisms observed in this experiment is in contrast to classic initiation models that assume that a temporally-constant fraction of saltating grains begin motion through either aerodynamic entrainment or splash (e.g. *Pomeroy and Gray*, 1990). As seen in Fig. 9 and the supplementary video, in intermittent conditions this variability is magnified, as splash regimes themselves are intermittent, and depend on sufficient wind speed for adequate particle acceleration upon

ejection from the snow surface.

## 5 Conclusion

This is the first investigation to measure outdoor snow particle flux and velocity immediately above the snow surface. It provides an opportunity to test certain observations of saltating particle flux trajectories measured in wind tunnels. Though observations were restricted to moderate wind speeds and

intermittent transport, they show the importance of creep to the initiation of blowing snow transport and the transition to full saltation. Being able to relate high frequency turbulent wind speed to snow transport (e.g. *Guala et al.,* 2008) is critical in the alpine environment (*Naaim-Bouvet et al.,* 2011) and this study makes a contribution to understanding these dynamics.

PTV has proven to be a viable avenue for exploring complex wind-snow interactions at millisecond

timescales in natural, non-steady state, high Reynolds number wind conditions. These results support the need for further conceptual advancement of models of snow particle movement, including initiation, rebound, multiple types of motion and the interaction of turbulent sweeps with particle erosion and entrainment. Over short timescales, snow particle motion is influenced by complex wind speed dependent initiation and rebound dynamics, including a dense near-surface flow whose presence and

variation cannot be described by scalar aerodynamic entrainment and splash parameterizations. Wind-to-snow and snow-to-snow momentum transfer in the first few mm above the surface is critical for

driving mechanisms of transport initiation and providing lower boundary conditions for two-phase atmospheric flows.

The results show a wide spectrum of particle motions exist with near-surface and upper level snow particle transport intrinsically linked through momentum and mass balances. The relative contributions of near-surface and upper level transport depend on wind strength and snowpack properties. Sand saltation velocity distribution models do not comprehensively describe transport of complex snow crystal structures such as the previously undescribed tumblon motions or snow particle shattering and sintering. Low-energy near-surface particles contributed significantly to snow transport as high near-surface particle concentrations compensated for reduced particle speeds, and sustained a layer of peak particle momentum and mass flux. The low-energy grains also contributed considerably to saltation by being a reservoir of particles bouncing into saltation and by breaking snow bed matrix bonds, thus making particles more available by reducing the wind drag required for splash and entrainment.

The ability of the snowpack surface to absorb wind and particle momentum in the dense near-surface region of particle transport appears variable and substantial. The role that low-energy near-surface particles play in the wind-snow momentum balance and mass flux appeared dependent on snow surface hardness. Wind-hardened surfaces shared several trends similar to that of rigid sand-beds, though not all wind-tunnel observations could be replicated. As saltation dynamics are dependent upon creep particle motions, which mediate exchange between the snow surface and blowing snow, creep dynamics changes over varying surface hardnesses also result in changes in saltation, such as changes in velocity gradients, particle concentration, and rebound dynamics. Therefore the near-surface snow transport has far more intricate dynamics and greater flux relevance than previously described.

In the present study, the near-surface particle velocities reflected instantaneous wind speed fluctuations and never achieved equilibrium. As the snow transport was in constant readjustment to changes in wind velocity, and short time scale turbulence characteristics did not scale with long time averages, further characterizations of the time scale of relevance or relevant turbulence scaling relations in alpine terrain need to be performed before a steady-state equilibrium type saltation model (e.g. *Pomeroy and Gray*, 1990; *Doorschot and Lehning,* 2002) can be deemed appropriate for these situations. Furthermore, as contributions of shear stress from different RS quadrants are spatially and temporally variable and the

mechanics of transport vary during gusts as seen in Section 3.3, the need to account for variable shear stress in high-resolution modeling (*Doorschot et al.,* 2004; *Groot Zwaaftink et al*., 2014) is reinforced.

It could be very useful to compare modeled entrainment and splash ratios with PTV datasets, however longer recording times over a larger variety of snow types would be necessary to obtain statistically

significant comparisons. Specifically, whether longer time-averaged statistics can account for periods of varying initiation during intermittent saltation, or can only apply in more nearly steady-state environments would be a useful finding for high temporal resolution applications (e.g. *Groot Zwaaftink et al.*, 2014). PTV shows potential to answer many open questions in blowing snow research through quantification of momentum redistribution in very near-surface particle motions. The use of high

temporal-resolution outdoor PTV measurements may prove useful in future work for understanding how turbulence influences blowing snow processes in natural settings.

## 6. ACKNOWLEDGEMENTS

The authors acknowledge funding from the Canadian Foundation for Innovation, the Natural Sciences and Engineering Research Council of Canada, the Changing Cold Regions Network, Canada Research

Chairs, the Global Institute for Water Security and Alberta Agriculture and Forestry.  The assistance of Nicolas Leroux and the Fortress Mountain Resort in logistics is gratefully noted, as are the suggestions from Florence Naaim-Bouvet to improve the manuscript. Data is available upon request directly from the authors, john.pomeroy@usask.ca.

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

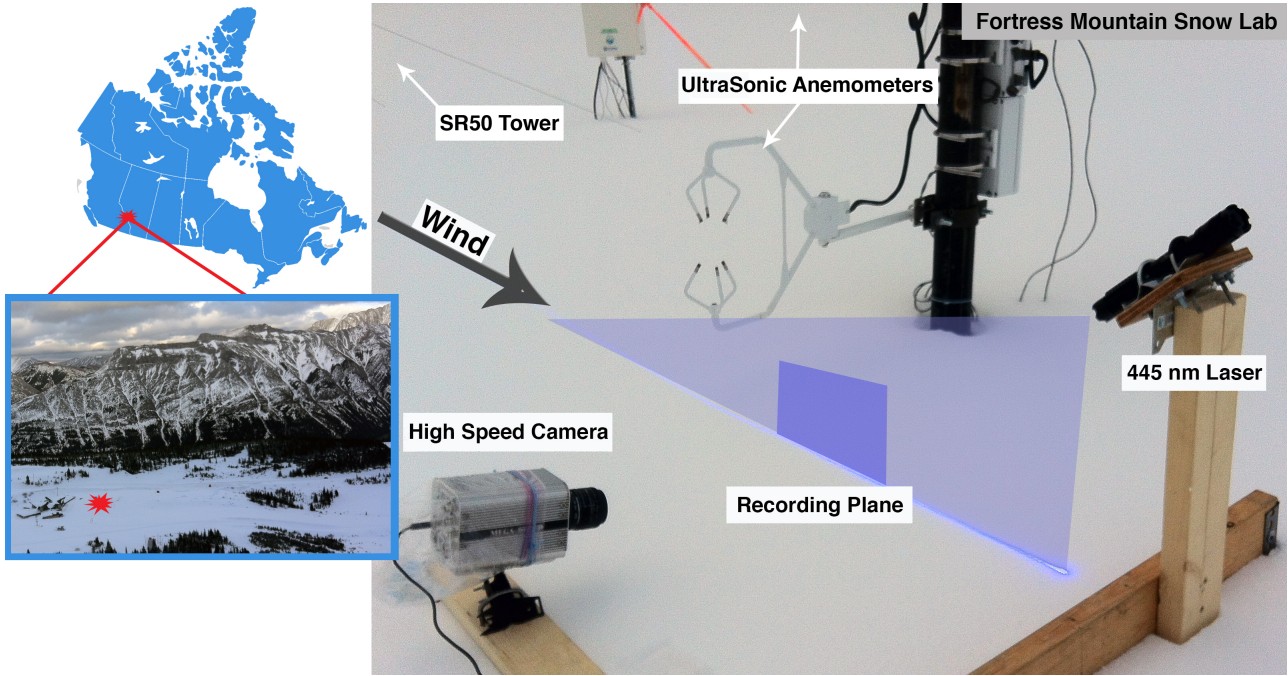

**Figure 1: Blowing snow instrument setup and location of field site, Fortress Mountain Snow Laboratory, Alberta, Canada**

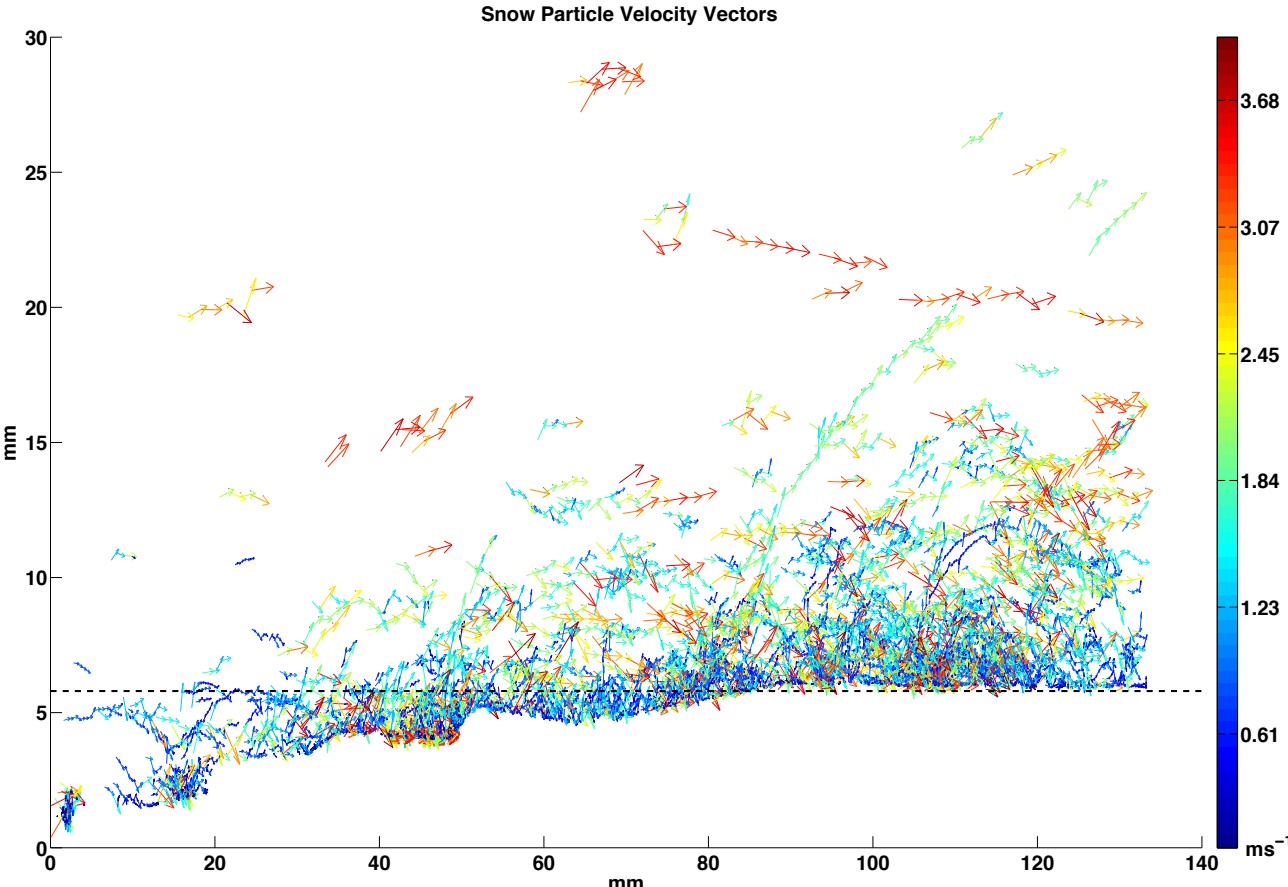

**Figure 2: Sparse snow particle velocity vector field during one second of recording on 23 March, vector colors scaled according to total particle speed. The dashed line shows reference below which particles are influenced by microtopography.**

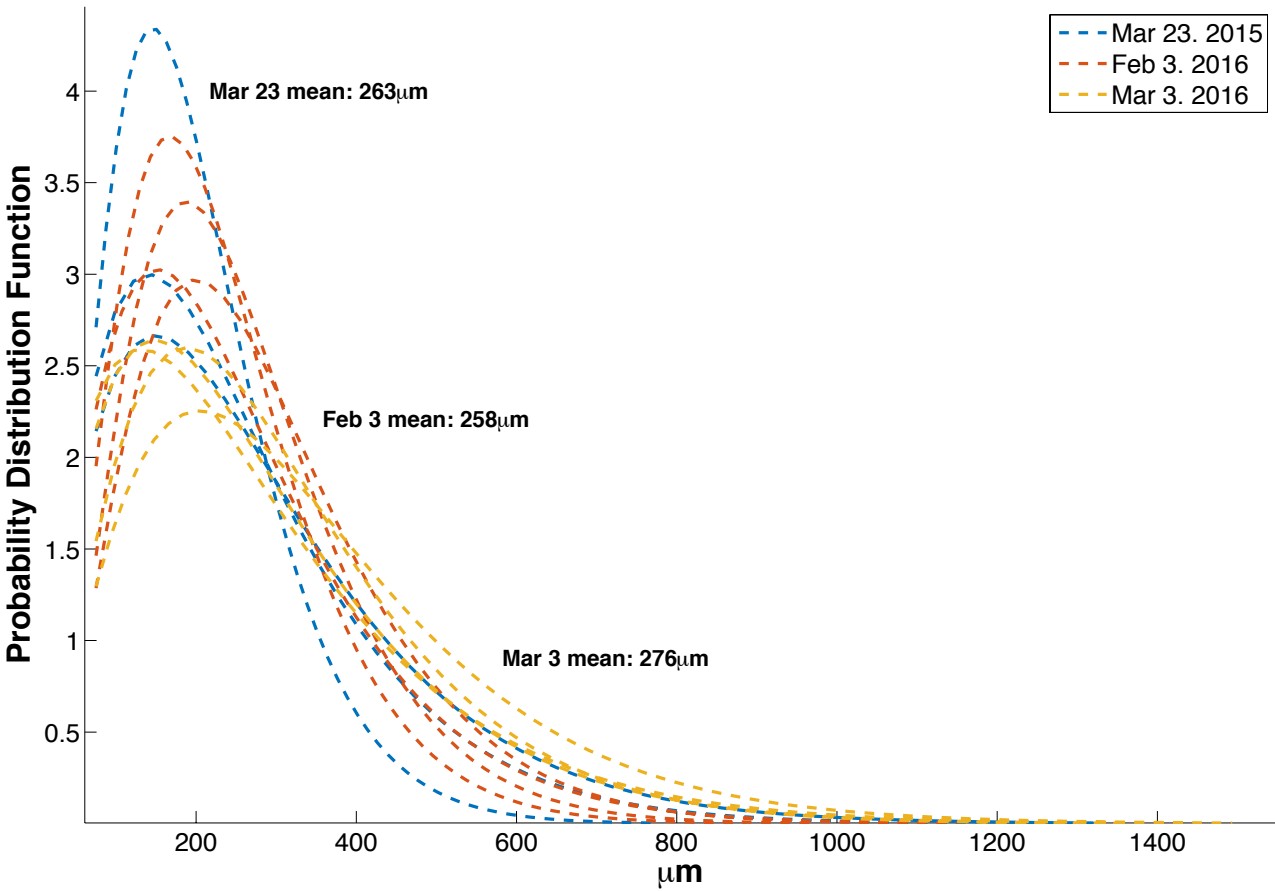

**Figure 3:** Two parameter Gamma distributions of particle diameters from each recording. Diameter measurements were obtained through black and white video binarization and equivalent diameter calculations of flood-fill identified connected components.

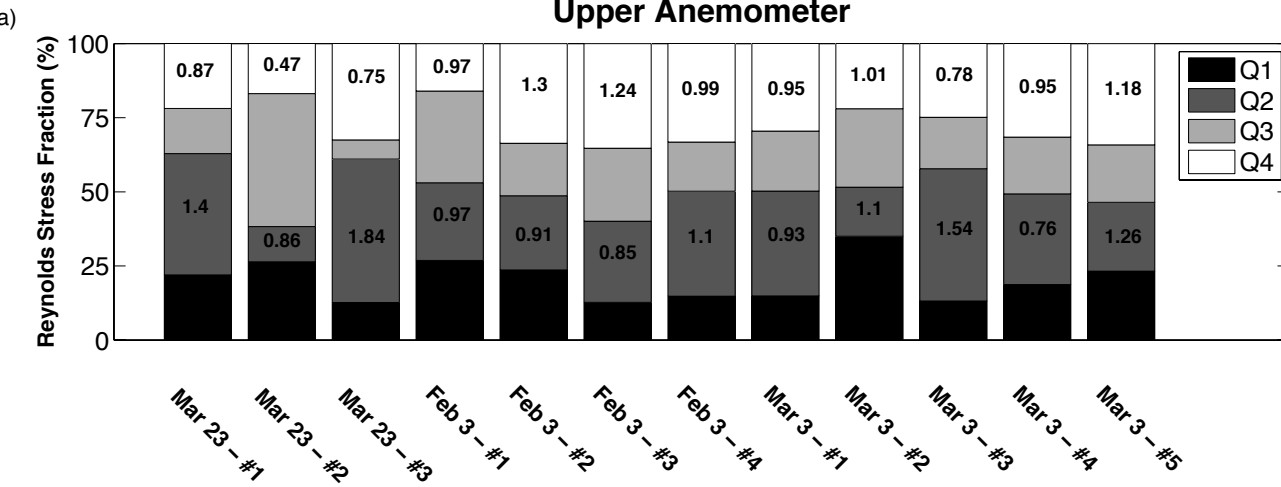

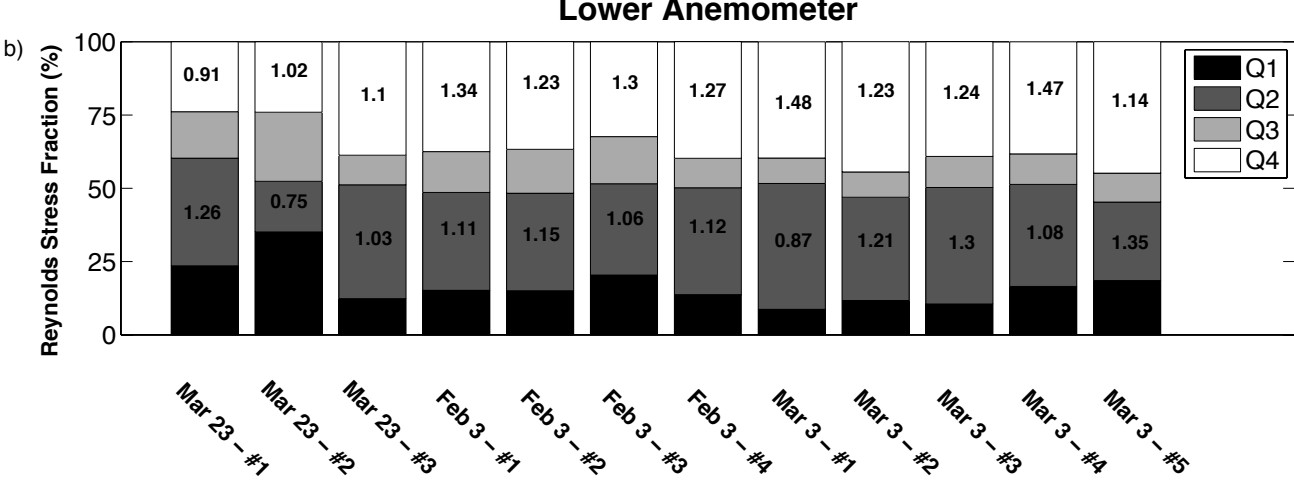

**Figure 4: Percentage of Reynolds Stress distributed by Quadrants Analysis during 12 Blowing Snow Recordings with impact factor (% Stress/% Time) inset in Q2 and Q4 events. Note dominance of Q2 and Q4 generated stress. Mar 23 - #2 Low anemometer measurement contaminated.**

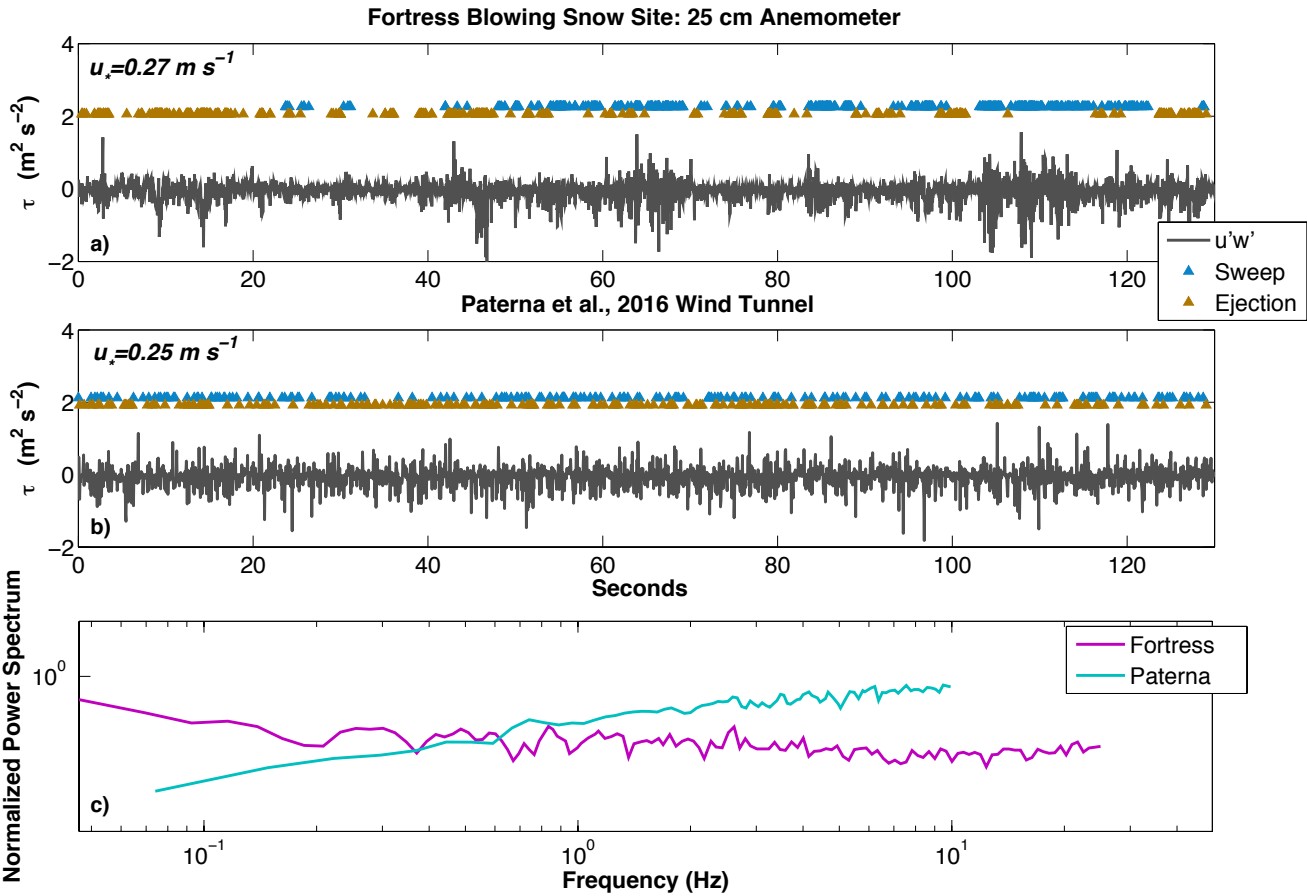

**Figure 5: Plots a & b compare Reynolds Shear Stress signals from 3 February 2016 and Paterna et al. (2016) wind tunnel blowing snow studies. Triangles indicated Sweep and Ejection events larger than one standard deviation of Reynolds Stress. C) Normalized Power Spectral Density for streamwise velocity for the two time series.**

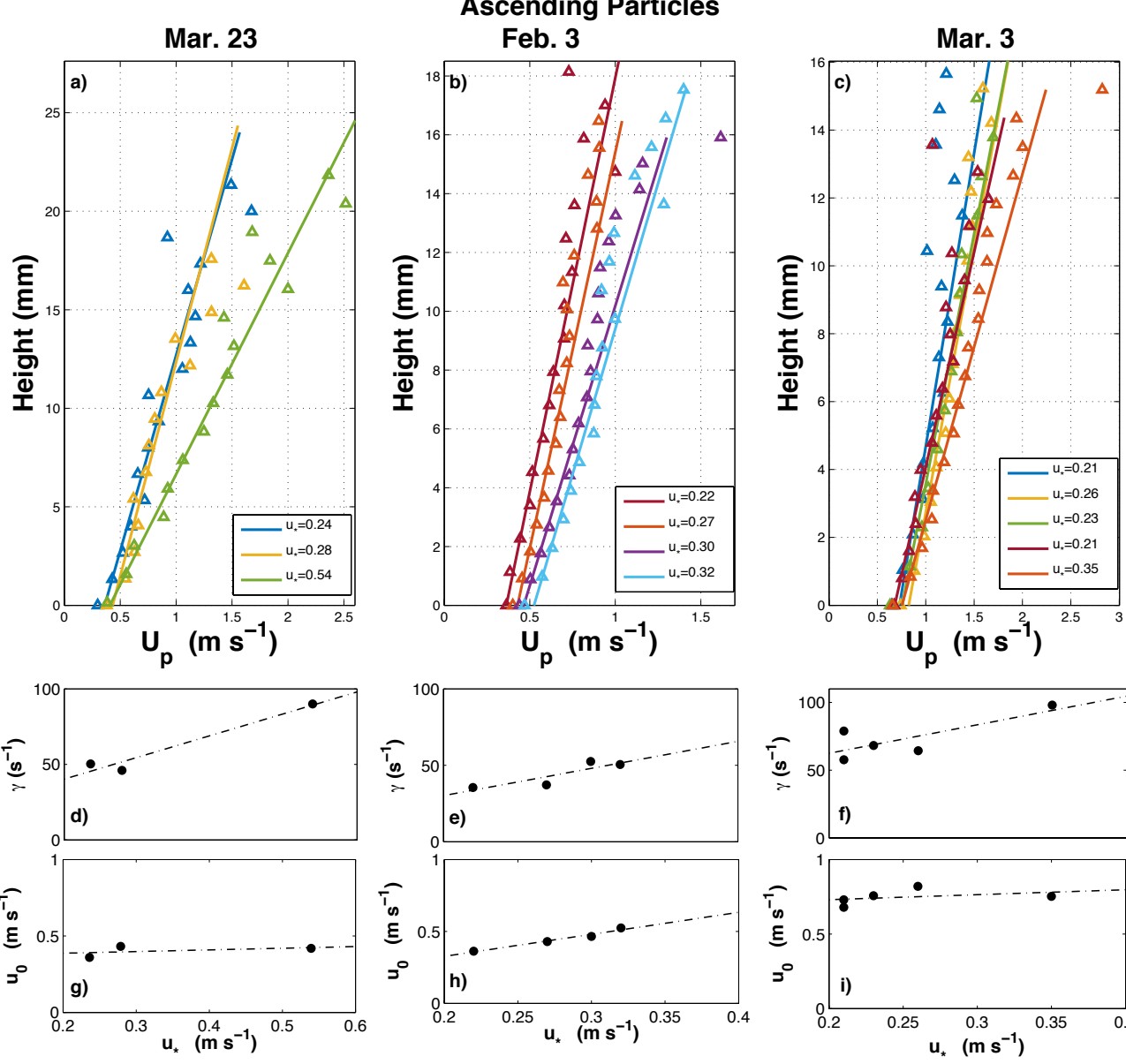

**Figure 6: a-c)** Average ascending snow particle horizontal velocities in lower saltation layer (triangles) with best fit linear profile for first 10 mm. $u_*$ values are calculated from 200 cm, 25 cm, and 10 cm for Mar 23, Feb 3, and Mar 3, respectively. **d-f)** Friction velocity versus particle velocity gradient ($\gamma$) for each recording. **g-i)** Friction velocity versus particle slip velocity ($u_0$) for each recording.

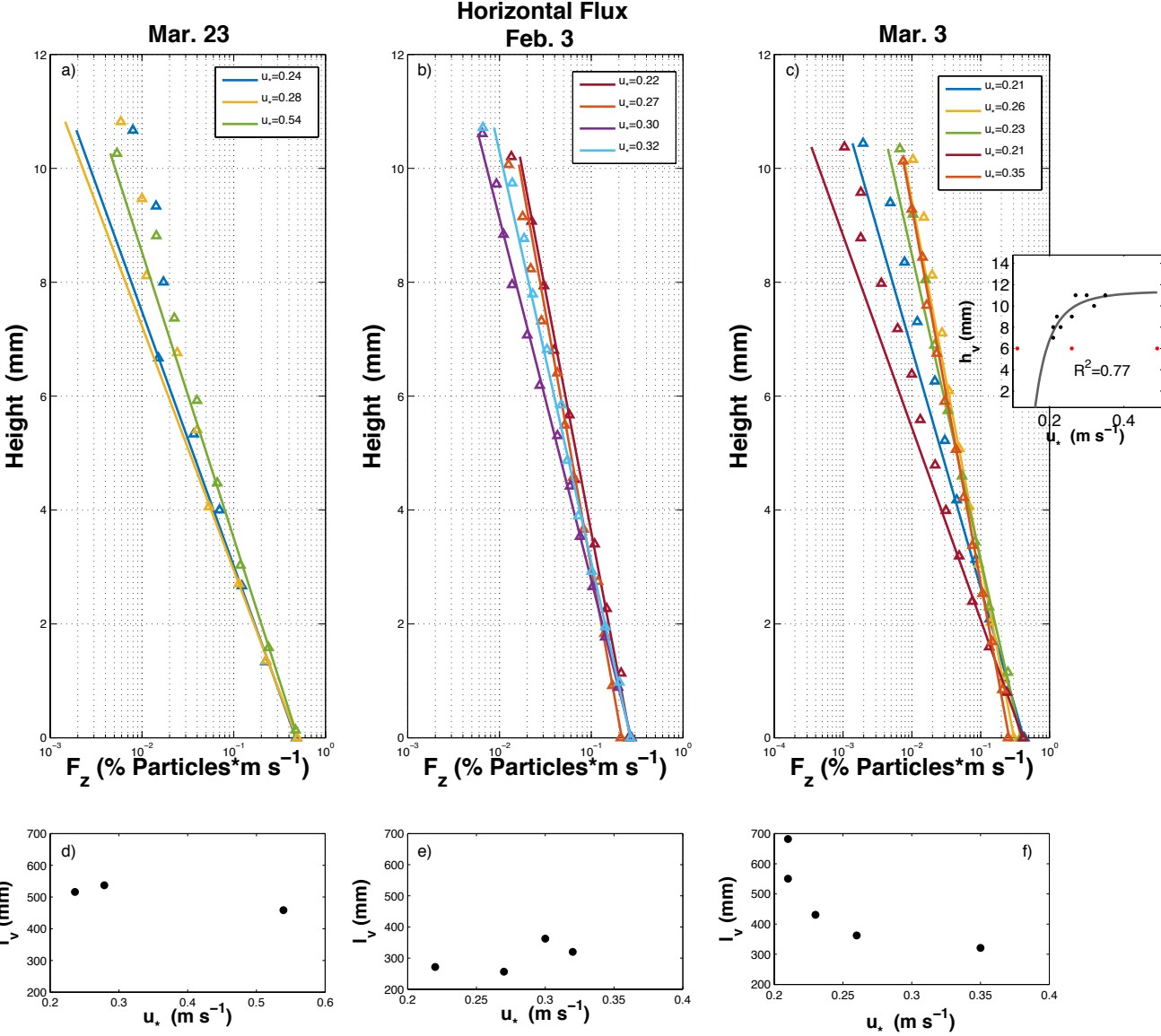

**Figure 7: Plots a-c are mean horizontal flux measurements ($F_z$) and best fit exponential decay. Plots d-f are friction velocity versus decay length for each night. Friction velocity versus $h_v$ (height below which 75% number flux occurred) for all nights with power law curve fitting is seen in the right inset. Values of $h_v$ from 23 March 2015 are marked in red.**

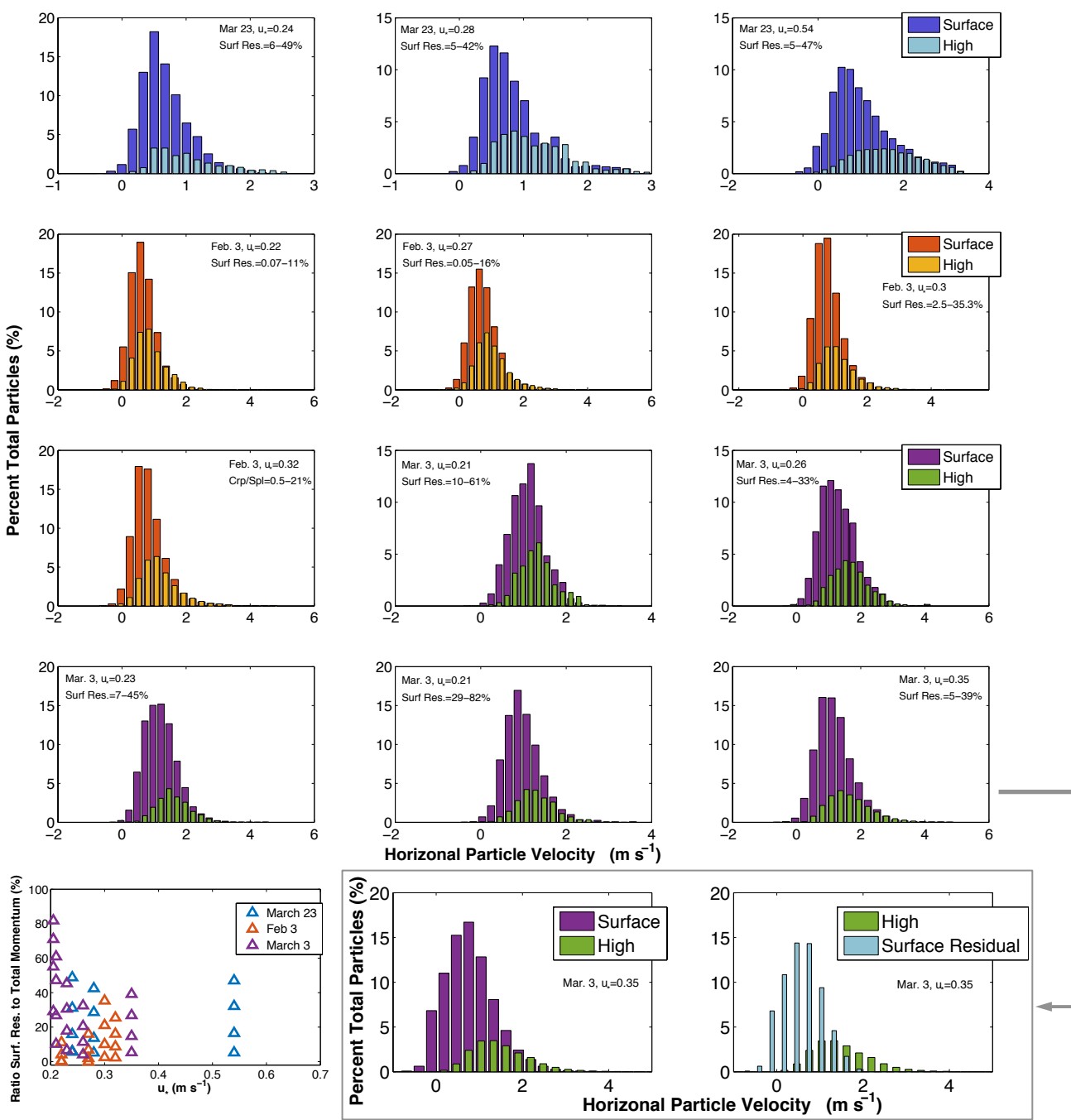

**Figure 8: Horizontal particle velocity histograms for near-surface and upper region descending particles for each recording over the three nights (each colored differently). Near-surface residual highlighted for Mar 3 #5. The two-region delineation was set at 3 mm for Mar 23 and Mar 3, and 4 mm for Feb 3. Bottom Left: Surface residual to total momentum ratios for each recording.**

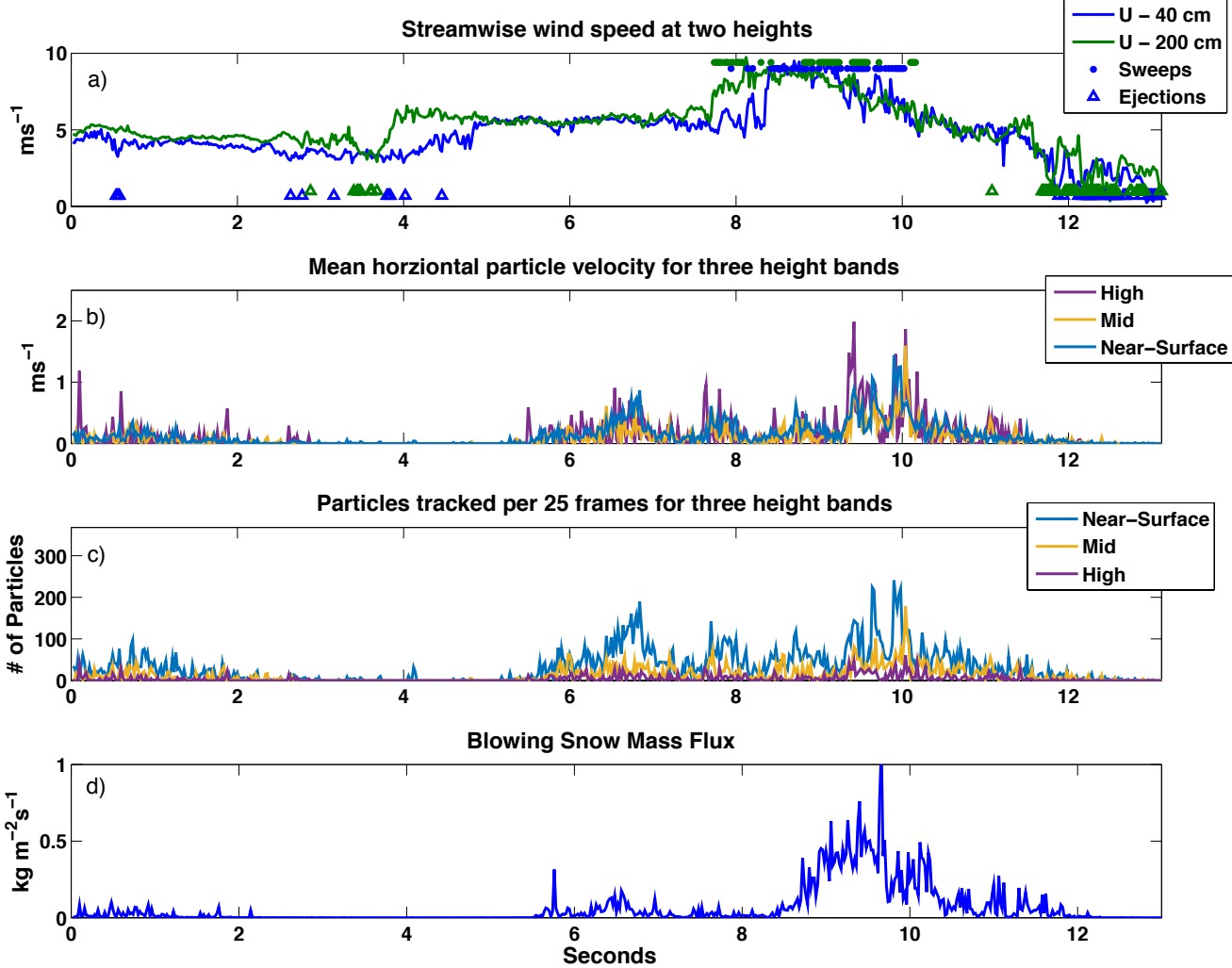

**Figure 9: 23 March, recording #3 time series. a) 50 Hz streamwise wind speed at 40 and 200 cm above snow surface. b) 50 Hz Snow particle velocities obtained by binning particle vectors in three height bands (1<z<4 mm (Near-Surface), 4<z<8 mm (Mid), 8<z<30 mm (High)), then temporally averaged over 25 frames. c) Number of tracked particles in same height bands per 25 frames, d) Instantaneous blowing snow flux rates $Q_s$ ($kg/m^2s$) (1250 Hz).**

| Date | Snow surface conditions and weather | Density of loose surface grains | 2 m Air/Snow Surface Temp | Surface Hardness (HHI) | Blowing Snow Grain Size ($\mu$m) |
|---|---|---|---|---|---|
| Mar. 23 | 5 cm graupel over old snow with light flurries | ~350* (kg m$^{-3}$) | -1C/(-)* | Fist over Melt-Freeze Crust (1 - 4) | 263 (Max: 1200) |
| Feb. 3 | Fine decomposing grains on windslab/sastrugi. No precip. | 228 (kg m$^{-3}$) | -10C/-10C | 1 Finger-Pencil (3 - 4) | 258 (Max: 850) |
| Mar. 3 | Fresh snow. No precip. | 156 (kg m$^{-3}$) | -2C/-5C | Fist (1) | 276 (Max: 2500) |

**Table 1: Descriptions of the snowpack for each night of recording including a description of the condition of the snow surface and concurrent precipitation, bulk density of the top 5 cm of grains, mean air temperature at the upper anemometer, snow surface temperature, hand surface hardness and HHI values following** *Fierz et al.* **(2009), as well as snow grain size as determined from the blowing snow video. \*Density and Snow Surface temperature not available on March 23 with density estimated from** *Pruppacher and Klett* **(1997).**

| Recording | Duration/Frames | $\bar{u}$ m s⁻¹ | $u_*$ m s⁻¹ | $z_0$ mm | $I$ % | $S$ (non-dim) | $Q_s$ kg m⁻² s⁻¹ |
|---|---|---|---|---|---|---|---|
| **23/3/15 #1** | 7.3 s/9147 | | | | | | 0.0146 |
| **200 cm** | | 1.2 (5.5) | 0.61 (0.23) | 910.1 (0.1) | 1.10 (0.22) | 0.169 (0.028) | |
| **40 cm** | | 1.1 (4.7) | 0.46 (0.26) | 126.9 (0.3) | 1.13 (0.27) | 0.096 (0.034) | |
| **23/3/15 #2** | 11.4 s/14299 | | | | | | 2.54e-4 |
| **200 cm** | | 1.9 (4.3) | 0.48 (0.22) | 410.6 (0.8) | 1.04 (0.24) | 0.139 (0.039) | |
| **40 cm** | | 1.7 (4.2) | 0.40 (0.08) | 69.9 (1.3 e-4) | 1.04 (0.16) | 0.096 (0.007) | |
| **23/3/15 #3** | 13.1 s/16404 | | | | | | 0.0237 |
| **200 cm** | | 1.4 (5.3) | 0.28 (0.55) | 312.2 (42.4) | 0.80 (0.40) | 0.037 (0.148) | |
| **40 cm** | | 1.3 (4.8) | 0.24 (0.57) | 116.5 (8.0) | 0.79 (0.45) | 0.027 (0.122) | |
| **3/2/16 #1** | 27.8 s/ 43075 | | | | | | 0.0179 |
| **155 cm** | | 4.67 (5.78) | 0.27 (0.26) | 1.7 (0.3) | 0.39 (0.22) | 0.038 (0.036) | |
| **25 cm** | | 3.96 (4.9) | 0.29(0.22) | 1.6 (0.06) | 0.41 (0.21) | 0.044 (0.026) | |
| **3/2/16 #2** | 28.0 s/ 57487 | | | | | | 0.0375 |
| **155 cm** | | 3.91 (5.35) | 0.22 (0.23) | 1.8 (0.4) | 0.45 (0.23) | 0.025 (0.027) | |
| **25 cm** | | 3.33 (4.54) | 0.22 (0.27) | 1.0 (0.2) | 0.46 (0.25) | 0.025 (0.037) | |
| **3/2/16 #3** | 28.1 s/ 57528 | | | | | | 0.0547 |
| **155 cm** | | 4.3 (5.7) | 0.34 (0.45) | 12.2 (12.5) | 0.44 (0.27) | 0.054 (0.106) | |
| **25 cm** | | 3.67 (4.7) | 0.27 (0.3) | 1.7 (0.7) | 0.44 (0.27) | 0.034 (0.045) | |
| **3/2/16 #4** | 28.0 s/ 57409 | | | | | | 0.044 |
| **155 cm** | | 3.1 (6.94) | 0.31 (0.33) | 35.4 (0.4) | 0.83 (0.17) | 0.056 (0.055) | |
| **25 cm** | | 2.6 (5.66) | 0.31 (0.33) | 14.7 (0.4) | 0.84 (0.2) | 0.056 (0.055) | |
| **3/3/16 #1** | 27.9 s/ 43182 | | | | | | 0.0046 |
| **140 cm** | | 5.33 (4.58) | 0.30 (0.16) | 1.7 (0.03) | 0.28 (0.16) | 0.035 (0.013) | |
| **10 cm** | | 4.3 (3.75) | 0.23(0.21) | 0.2 (0.3) | 0.29 (0.19) | 0.021 (0.021) | |
| **3/3/16 #2** | 27.9 s/ 43182 | | | | | | 0.1474 |
| **140 cm** | | 4.33 (5.47) | 0.25 (0.42) | 1.8 (10) | 0.3 (0.23) | 0.030 (0.084) | |
| **10 cm** | | 3.57 (4.33) | 0.20 (0.26) | 0.3 (0.1) | 0.3 (0.23) | 0.019 (0.033) | |
| **3/3/16 #3** | 27.9 s/ 43182 | | | | | | 0.1561 |
| **140 cm** | | 4.7 (5.5) | 0.27 (0.23) | 2 (0.04) | 0.26 (0.16) | 0.035 (0.02) | |
| **10 cm** | | 3.85 (4.6) | 0.22 (0.20) | 0.4 (0.1) | 0.27 (0.18) | 0.023 (0.026) | |
| **3/3/16 #4** | 28.1 s/ 57528 | | | | | | 0.0542 |
| **140 cm** | | 4.6 (5.54) | 0.58 (0.24) | 82 (0.2) | 0.53 (0.17) | 0.149 (0.028) | |
| **10 cm** | | 3.7 (4.35) | 0.45 (0.21) | 15 (0.9) | 0.54 (0.2) | 0.09 (0.021) | |
| **3/3/16 #5** | 28.1 s/ 57528 | | | | | | 0.3467 |
| **140 cm** | | 4.3 (6.06) | 0.50 (0.31) | 63 (0.8) | 0.61 (0.36) | 0.131 (0.046) | |
| **10 cm** | | 3.5 (4.73) | 0.42 (0.35) | 14 (2) | 0.63 (0.4) | 0.092 (0.06) | |

**Table 2: Wind characteristics: mean wind speed $\bar{u}$, friction velocity $u_*$, aerodynamic roughness height $z_0$, turbulence intensity $I$, and Shields Parameter $S$ for recordings on Mar. 23, 2015, Feb. 3 and Mar. 3, 2016. Average values from the 15 minutes surrounding each recording (and recording-only period in parentheses) are shown for the two measurement heights. Estimates of blowing snow flux $Q_s$ in $kg\ m^{-2}\ s^{-1}$ are included for recording only periods.**