# Peer review of "Near-Surface Snow Particle Dynamics from Particle Tracking Velocimetry and Turbulence Measurements during Alpine Blowing Snow Storms"

_The Cryosphere, 2016_

## Referee Comment (RC1) · Anonymous Referee #1 · 23 Jun 2016

This manuscript introduces the measurements of the blowing snow with the Particle Tracking Velocimetry (PTV) and tries to investigate the particle motions near the snow surface. I appreciate very much for the author's effort to apply the PTV technique in the field and successful observations. The attitude should be highly evaluated. However, as far as I read through the manuscript, I have got the impression that the data shown here is not always valuable for both the drifting and blowing snow research community, in particular, for the accurate modeling. PTV recordings shown in Figure 2 probably involve the meaningful information, however, I am afraid the surface bed is far from flat and the height difference amounted to 5 mm. It gives the substantial effect on both

particle speeds and the wind velocities shown in Figure 3. It is quite plausible that the reason why the wind speed showed the maximum at 2 to 7 mm and why a zero wind velocity zone above the saltation layer exists is related to the bed surface undulation. Bagnold's focal point is hard to refer under such conditions. Further, the friction velocity of 0.08 m/s at Rec. #2 is extremely low; under such conditions the blowing snow, never breaks out and keeps going, even though we take into account the turbulence effect. On the contrary, at Rec. #3, u* is extremely high. Under such conditions, I suppose the particle concentration near the surface increases largely and it makes hard to distinguish individual particle, that is, no precise particle tracking is available. In fact, I cannot agree with you more that the turbulences including ejection and sweep intermittent structures are key factors to initiate the snow particle motion, rather than the time averaged friction velocities, not only in the mountain area but on the flat snow surface. However, when you would like to set your focus on these issues, you need more specific and detailed analysis based on the high frequency data. Although the quadrant analysis has been tried, the explanation of the outcome is superficial, more detailed analysis related to the particle motion are essential. Presumably 1m distance between the PTV and the anemometer makes the quantitative comparison difficult? Further, the descriptions in the discussion and conclusion parts are mostly qualitative and look nothing but a pile of well known and predictable issues. More firm conclusions based on the more quantitative analysis are essential. I strongly recommend, first of all, the authors to reexamine the obtained data again and uncover the hiding useful ones. Further, an accumulation of more data will be preferable; in actual, saltation of 400 to 1000ïA■m diameter graupel is a very exceptional case. I cannot believe that such large particles kept saltation under the friction velocity of 0.08 m/s at Rec. #2r. At this stage, I regret to say that this manuscript has not been matured for the TC publication.

---

## Referee Comment (RC2) · Anonymous Referee #2 · 29 Jun 2016

General points

In this paper , the authors describe a set of innovative data acquired by Particle trecking Velocimetry during blowing snow events. Even if Particle Trecking Velocimetry is a classic wind-tunnel method for studying drifting particles, this is the first investigation to measure outdoor snow particle flux and velocity. It is really challenging and interesting and is a potential source of new knowledge. Such experiments should be made known to the scientific community. But even if the paper is potentially interesting, it seems that it is not suited for publication in its current state. I have several suggestions for

improvement if the authors would like to resubmit the manuscript.

- (1) The results are not described in sufficient details and conclusions are often not supported through the material presented. The authors must be aware of what information the data can provide and what they cannot provide. It is a persistent problem throughout the whole text. Sometimes, the new findings highlighted in the paper seem questionable.

- (2) It is surprising that the results are not discussed taking into consideration key measurement uncertainties (related to the position of surface bed and to the distance between PTV measurements and ultrasonic anemometer). This way, analysis could be enhanced.

- (3) The time series need to be extended, as –apparently- there are much more data available. However, it would be useful to know how many other cases (if any) could have been selected and why they were not presented here. I would encourage the authors to present more of the valuable data. If not, the research paper must be considered as a Brief communication (http://www.the-cryosphere.net/about/manuscript_types.html)

- (4) The text could be more concise and focused. Similar points are discussed in several places of the text.

- (5) Moreover the nature of discussion should be more quantitative than qualitative.

- (6) Papers supporting the reasoning should be properly referenced and used. Otherwise, it puts a doubt into readers' minds.

Example of specific points illustrating general points

- relating to the item 1

P7-L10 : From figures 3a-g, it is not evident that the constant particle velocity gradient is limited to a height below 10 mm, at least for me. Moreover it seems that the velocity

gradient estimated by linear regression includes all measurements points ? Is it the case ? How do the authors estimate that this 10 mm transition corresponds with the upper extent of the low-energy population ?

P8-L14 : The 10 mm threshold delimitating variances of blowing snow particles is not clear again. The decline in variances is not so pronounced. For sure it is difficult to directly estimate this value from the graph. Some orders of magnitude could be useful for the readers.

P9-L4 : I did not understand how the results help to illuminate the shift in snow transport mechanics when transitioning to particles in the tail of particles velocity distributions. The authors are expected to provide more explicit demonstration.

P10 : The authors have to explain in details how concurrent streamwise wind measurements show penetration of a turbulent sweep and why it is best to base our reasoning on streamwise wind measurements instead of Reynolds stress measurements.

P10-L4 (and figure 4) : New threshold values (4 mm and 8 mm) are given in this paragraph. Without any additional measurements, I see absolutely no reason why the authors change their tune. The authors previously compared the 10 mm threshold with values obtained by Ho et al., 2014, which correspond to the limit between saltation and reptation (particles are divided in two populations on the basis whether or not they rise high enough to be affected by the flow strength). On which basis (other than a qualitative approach for one blowing snow event) do the authors decide that 4 mm is the limit between creep and saltation ? It is really confusing.

P9-L16 : How do the authors consider that particles are in creep ? by the position of the particles (i.e a particle seen below 4 mm is considered as being in creep ? ) If at a given time a particle is at this position, it doesn't mean that later it will not be able to rise high enough to be affected by the flow strength.

-relating to the item 3

Neutral stability did not occur during the field campaign as it can be seen on table 1 (the Reynolds stress is not constant with the height). It is a pity because neutral conditions are quite usual on the site (80% of the 192 hourly periods studied in Helgason and Pomeroy, 2005). So, authors can't calculate the aerodynamic roughness which becomes a function of height. What are the Richardson numbers for these experiments ? Is there any other data under neutral stability over the course of the campaign ?

For recording 2 there is a strong difference between u* estimated by eddy covariance method at a height of 200 cm and a height of 40 cm. Is it possible for the blowing particles to disturb the measurements ? What are the drifting snow fluxes measured during recording-only period and during the 15 minutes surrounding each recording. There are quite unusual results which need to be commented (for example high value of roughness which can be smaller when estimating by flux-profile estimation techniques suggesting that the mean wind profile was in equilibrium with the snow surface, however the turbulence was not (Helgason and Pomeroy, 2005).

-relating to the item 4

P7-L10/20 : Paragraph about Ascending particles

P7-L21/P8-L8 : Paragraph about Descending particles

P8-L9/P8-L14: Paragraph about Ascending particles

It is a little bit confusing for the reader

-relating to the item 5

Figure 5 by itself is not an evidence that tumblons eroded many smaller crystals from the surface or shattered themselves and immediately became saltating grains, depending on impact velocity. Where are the measurements to show the effect of impact velocity ? Moreover impacting particles may travel transverse to the plane of light and may not be included on the second image. Conclusions must be based on a statistical approach.

-relating to the item 6

P3- L7: Ho et al., 2011 does not address grain velocity distribution functions

P8-L27: Ho et al., 2012 deals with Particle velocity distribution in saltation transport. So when speaking about number density, the authors have to use the right reference (It is probably Ho et al., 2011). Ho et al., 2011 explaine that the particle volume fraction decreases with height at a given exponential rate in saltation layer. If the authors want to compare their results with Ho et al., 2011, they have to limit the analysis to the first centimeter and to draw the result in the same manner as Ho et al., 2011 (figure 8) with an inset including the characteristic decay length. Moreover the authors base their analysis on the fractional particle number flux whereas Ho et al., 2011 base their analysis on the particle volume fraction. If both results are compared, the authors have to take into account the volume of particles which can vary according to the wind speed.

P2-L29: Schmidt (1980) instead of Schmidt(1986)

P11-L18/24: Sigiura and Maeno, 2000 made a distinction between horizontal restitution coefficient and vertical coefficient restitution which are different from the restitution coefficient calculated from the authors. Moreover the calculation method completely differs. The authors should make it clear.

P7-L19: What is the numerical value of the transition height obtained by Ho et al., 2014 (2 zf ?) ? As far I can see from Figure 3 the Bagnold focus point zf is around 8 mm. Considering uncertainties in relation to the choice of 10 mm threshold both values are close together. What are the Shield numbers of the snow particles in the experiments ? Ho et al., 2014 remain that the results have been obtained in a finite range of Shields number from 0.04 to 0.2.

---

## Author Response (AR1)

**Reviewer #1 Comments and Response:**

This manuscript introduces the measurements of the blowing snow with the Particle Tracking Velocimetry (PTV) and tries to investigate the particle motions near the snow surface. I appreciate very much for the author's effort to apply the PTV technique in the field and successful observations. The attitude should be highly evaluated.

*Thank you.*

However, as far as I read through the manuscript, I have got the impression that the data shown here is not always valuable for both the drifting and blowing snow research community, in particular, for the accurate modeling.

PTV recordings shown in Figure 2 probably involve the meaningful information, however, I am afraid the surface bed is far from flat and the height difference amounted to 5 mm. It gives the substantial effect on both particle speeds and the wind velocities shown in Figure 3.

*Yes, we note the effect of surface "microtopography" on obtaining useable statistics is shown below the dashed line of influence. In the revised paper, we have introduced terrain-following coordinates (see response to Reviewer Comments #2). However, 5 mm rise over 130 mm equates to a little more than 2 degrees of slope. When we are considering blowing snow in alpine terrain, this is quite flat.*

*(Page 6, 2$^{nd}$ paragraph.)*

It is quite plausible that the reason why the wind speed showed the maximum at 2 to 7 mm

*We do not see this in our data.*

and why a zero wind velocity zone above the saltation layer exists is related to the bed surface undulation.

*A modeled zero wind velocity zone above in figure 3e-f could indeed be related to surface roughness. The issue at hand in the discussion is how in natural conditions, the commonly used log-law will predict erroneous wind profiles. Thus the choice of timescales for mean values and assumptions required for predicting wind speed for blowing snow in complex terrain must be of higher concern.*

*(Section 3.1)*

Bagnold's focal point is hard to refer under such conditions. Further, the friction velocity of 0.08 m/s at Rec. #2 is extremely low; under such conditions the blowing snow, never breaks out and keeps going, even though we take into account the turbulence effect.

*The wind data for Rec #2 has been compromised during the recording specific period, hence why the*

*difference between 2m and 40cm values is so much larger than during other recordings. This has been highlighted in the revised text.*

*(Page 9, line 17).*

5  On the contrary, at Rec. #3, u* is extremely high. Under such conditions, I suppose the particle concentration near the surface increases largely and it makes hard to distinguish individual particle, that is, no precise particle tracking is available.

*True, particle tracking does become difficult and this was mentioned in the discussion of figure 4 – (old P10 L24, P11 L4.)*

10  *(Page 8, line 23, Page 13, line 8. )*

In fact, I cannot agree with you more that the turbulences including ejection and sweep intermittent structures are key factors to initiate the snow particle motion, rather than the time averaged friction velocities, not only in the mountain area but on the flat snow surface.

However, when you would like to set your focus on these issues, you need more specific and detailed
15  analysis based on the high frequency data. Although the quadrant analysis has been tried, the explanation of the outcome is superficial, more detailed analysis related to the particle motion are essential.

*Thank you. We have expanded this analysis as well as the amount of data investigated in the revised paper.*

20  *(Section 3.1 and Section 3.3)*

Presumably 1m distance between the PTV and the anemometer makes the quantitative comparison difficult?

*This distance has a considerably smaller effect in the atmospheric surface layer than in wind tunnels because the eddies driving the turbulent structures are much larger. This lateral distance was chosen to*
25  *preclude any bluff-body influence of the camera/laser apparatus. Subsequent data has been included with a smaller distance between PTV and anemometer in a spanwise orientation. Please refer to the response to Reviewer #2 Comments.*

*(Page 5, 1ˢᵗ paragraph)*

Further, the descriptions in the discussion and conclusion parts are mostly qualitative and look nothing
30  but a pile of well-known and predictable issues.

*We have obtained many similar results to aeolian transport studies in much more controlled wind*

*tunnel environments, this shows how the PTV approach in natural conditions can confirm certain results found in more controlled, but less natural wind tunnel environments. The scientific blowing snow community would benefit from the knowledge that such techniques may be used in nature, as the potential gains from employing these techniques outdoors are considerable, albeit with further quantitative results coming only after "well-known and predictable issues" are first addressed.*

More firm conclusions based on the more quantitative analysis are essential. I strongly recommend, first of all, the authors to reexamine the obtained data again and uncover the hiding useful ones.

*Data from two more nights of recording have been included in the analysis and discussion in the revised paper. Consideration of these data and further analysis of other data has led to stronger conclusions.*

Further, an accumulation of more data will be preferable; in actual, saltation of 400 to 1000µm diameter graupel is a very exceptional case. I cannot believe that such large particles kept saltation under the friction velocity of 0.08 m/s at Rec. #2r.

*Addressed above. Improved particle diameter algorithms have been used with particle size distributions included in the Document Supplement Figure 1, as well as mean and max values in Table 1.*

**Reviewer #2 Comments and Response:**

In this paper, the authors describe a set of innovative data acquired by Particle trecking Velocimetry during blowing snow events. Even if Particle Trecking Velocimetry is a classic wind-tunnel method for studying drifting particles, this is the first investigation to measure outdoor snow particle flux and velocity. It is really challenging and interesting and is a potential source of new knowledge. Such experiments should be made known to the scientific community. But even if the paper is potentially interesting, it seems that it is not suited for publication in its current state. I have several suggestions for improvement if the authors would like to resubmit the manuscript.

**- (1) The results are not described in sufficient details and conclusions are often not supported through the material presented. The authors must be aware of what information the data can provide and what they cannot provide. It is a persistent problem throughout the whole text. Sometimes, the new findings highlighted in the paper seem questionable.**

*We have expanded the discussion of results, added new data to back this discussion and included new results that are strongly backed by analysis of this data. We have clarified the findings from the data and removed speculative discussion or identified it more clearly in the discussion section. We are very confident in the veracity and certainty of our results and note that as these come from outdoor measurements, they are more likely to be representative of natural blowing snow than are wind tunnel simulations under less turbulent and more idealized two phase flow conditions.*

P7-L10 : From figures 3a-g, it is not evident that the constant particle velocity gradient is limited to a height below 10 mm, at least for me.

*After consideration, we agree with this comment and so have expanded our analysis with new data and now Figure 3 has been restructured using terrain following coordinates. Now, particle velocity profiles of 12 recordings (9 new) over three nights of recordings are presented encompassing 283 seconds and 513,000 frames. From these we have found that the horizontal velocity profiles of ascending particles start to diverge from linear plots above heights that vary from 8 to 15 mm.*

Moreover it seems that the velocity gradient estimated by linear regression includes all measurements points? Is it the case?

*In the new figure 3, the velocity gradient has been estimated using particles in the first 10 mm of the terrain following coordinate system.*

How do the authors estimate that this 10 mm transition corresponds with the upper extent of the low-energy population?

*As noted above we no longer conclude that there is a 10 mm threshold (pg. 12 l. 5-10, ). Aeolian transport exhibits a continuous spectrum of particle velocities and so the gradient of motions*

*from "low-energy" to "high-energy" does not contain a discrete step. Using the Ho et al. (2014) definition, derived for saltating sand in a wind tunnel, where the two populations are distinguishable if grains are either responsive or irresponsive to changes in 'wind strength,' we found that nearly all snow particles outdoors are responsive to 'wind strength'. This manuscript has been restructured to discuss this continuum of motion through examination of particle velocity distributions (as in Ho et al. 2012) and identifying the region of transition from a creep dominated particle population to higher energy grains. This highlights the important role of creep in forming the lower boundary condition and "source" for saltation.*

P8-L14: The 10 mm threshold delimitating variances of blowing snow particles is not clear again. The decline in variances is not so pronounced. For sure it is difficult to directly estimate this value from the graph. Some orders of magnitude could be useful for the readers.

*We agree. The variance argument has been eliminated and replaced by data in a new Figure 3.*

P9-L4 : I did not understand how the results help to illuminate the shift in snow transport mechanics when transitioning to particles in the tail of particles velocity distributions. The authors are expected to provide more explicit demonstration.

*This is now explicitly demonstrated as follows: This and other studies have shown that the density of flow decreases exponentially with height. The two particle velocity histograms (above and below 4mm) indicated at what velocities the majority of particles were moving in each height region. 4 mm was an arbitrary divide to separate near surface and above surface flow regions. Very close to the surface in the densest part of the flow, the flux was dominated by slow moving ("creep") particles, but also with a contribution of relatively fewer high-energy ("saltation") particles that has not been described before. Above the snow surface, there was a continuum of particle velocities that was dominated by high-energy particles. The increasing proportion of high-energy particles with distance from the surface is due to the need for a greater velocity to reach greater heights on a ballistic trajectory from the surface and the subjection to stronger winds with increasing distance above the surface.*
*We were illuminating the point that high energy particles are also counted at the <4mm level as well. Therefore typical saltation measurements that limit data above the creep layer are missing critical data, whereas, near-surface measurements include both populations. Additionally, there is an intrinsic coupling of low-energy ("creep") population with high-energy ("saltation") particles as they both occupy the same spatial location, and the creep layer feeds upper regions of saltation as mentioned in the new discussion of Figure 4 (Figure 9).*
*(Pg 15-16.)*

P10 : The authors have to explain in details how concurrent streamwise wind measurements show penetration of a turbulent sweep.

*Figure 4 has now been adapted (Figure 8) to show time periods of significant ejection and*

*sweep using Quadrant Analysis with a hole size of 1. There is an ejection (triangles) that precedes the main sweep (dots) event at 8 seconds. Sweeps by definition indicate positive fluctuations in streamwise wind speed with negative fluctuations in the vertical. As the sweep first became evident at 2 m and then at 0.4 m height, this motion clearly penetrated from above.*

5  *The discussion in the manuscript discusses these issues at length now,*
*(Section 3.3, pg 17 line 19, pg 18 line 19.)*

Why is it best to base our reasoning on streamwise wind measurements instead of Reynolds stress measurements?

*This is an excellent question. This empirical finding from our data agrees with conclusions from other field and wind tunnel sand studies (Bauer et al., 1998; Sterk et al., 1998; Schonfeldt and von Lowis 2003, Leenders 2005). Strong ejections (high Reynolds stress) events at 2-4 second, and >12 seconds did not result in blowing snow transport. Instead it was the positive fluctuations in streamwise*
15  *wind speed that resulted in flux. The nature of turbulence near the surface has an influence of transport initiation and a simple Reynolds stress cannot fully describe this, whilst near surface wind speeds can better reflect sweeps and ejection events. Further analysis of this phenomenon is of interest but would be a very extensive addition to this paper and so we now call for it in the conclusions and suggest it for future research.*
20  *(Pg 11 line 3, pg 23)*

P10-L4 (and figure 4): New threshold values (4 mm and 8 mm) are given in this paragraph. Without any additional measurements, I see absolutely no reason why the authors change their tune. The authors previously compared the 10 mm threshold with values obtained by Ho et al., 2014, which correspond to
25  the limit between saltation and reptation (particles are divided in two populations on the basis whether or not they rise high enough to be affected by the flow strength).

*These three arbitrary heights are not thresholds, they were chosen to illuminate the subtle differences in particle transport and the continuum of motion as grains begin motion and begin*
30  *bouncing to greater heights as wind speeds increase. They do not indicate hard thresholds of "creep" versus "saltation" regimes.*
*This will be reflected more clearly in the text to avoid such misinterpretations. In the revised text we have adopted the Ho et al. (2012) probability distribution explanation of "long tails" corresponding to the population of high energy grains.*
35  *(Pg 4, line 2, 5, 9-10. Pg 15, line 12 to Pg 16 line 5. Pg 17, line 26, pg 20, line 26, Page 21 line 6)*

On which basis (other than a qualitative approach for one blowing snow event) do the authors decide that 4 mm is the limit between creep and saltation? It is really confusing.

40  *This arbitrary height is not a threshold, it was chosen to illuminate the subtle differences in particle transport and the continuum of motion as grains begin motion and begin bouncing to greater heights as wind speeds increase. To clarify, we have revised the text to more clearly show a transition*

*region between the "creep" dominated and "full saltation" dominated flow regions. This was already mentioned several times in the text. The emphasis is that the creep layer contributes substantially to blowing snow flux and is the source of saltation.*
*(Pg 4, line 2, 5, 9-10. Pg 15, line 12 to Pg 16 line 5. Pg 17, line 26, pg 20, line 26, Page 21 line 6*
5   *Pg 12, line 11.)*

P9-L16 : How do the authors consider that particles are in creep? by the position of the particles (i.e a particle seen below 4 mm is considered as being in creep ? ) If at a given time a particle is at this position, it doesn't mean that later it will not be able to rise high enough to be affected by the flow
10   strength.

*As explained above, we do not use a height threshold, but examine particle speed and consider a continuum of motions between classical creep and saltation. The histogram in figure 3 (now Figure 7) was designed for this purpose: low-energy particles near the surface are more likely to be "creep".*
15   *(page 9 second paragraph). The difference line on P9 – L16 identifies precisely the low-energy particles present near the surface, whereas the high-energy particles that are present at both heights are not counted. This is emphasized clearly in the revised paper.*
*(Pg 16, line 7.)*

20   **(2) It is surprising that the results are not discussed taking into consideration key measurement uncertainties (related to the position of surface bed and to the distance between PTV measurements and ultrasonic anemometer). This way, analysis could be enhanced.**

*Natural blowing snow as found outdoors has a naturally uneven bed – in contrast to its common*
25   *representation in wind tunnel studies. Uncertainty with respect to the position of the bed is briefly addressed with respect to the line indicating "influence of microtopography" in figure (2, 3). The issue of surface influence is now considered quantitatively by using a terrain-following coordinate system.*
*(Page 6 2nd paragraph)*

30   *Measurement uncertainties with respect to the distance between PTV and ultrasonic wind measurements can be considered negligible for time-averaged quantities as they are occurring at similar heights. Assuming Taylor's frozen turbulence hypothesis, and utilizing the mean wind speed during recordings, a lag time of approximately 1/3 of a second could be expected, much shorter than the duration of any given recording. For instantaneous measurement comparisons, these lag times are*
35   *discussed in that section.*
*The February 3 and March 3 laser measurements were made at closer distances (0.5 and 0.33 m, respectively) to the CSATs, perpendicular to the direction of wind flow. The effect of this orientation on comparing turbulence to transport measurements is mentioned in the revised text.*
*(Page 5, 1st paragraph . Page 23, line 11.)*

**- (3) The time series need to be extended, as –apparently- there are much more data available. However, it would be useful to know how many other cases (if any) could have been selected and**

**why they were not presented here. I would encourage the authors to present more of the valuable data. If not, the research paper must be considered as a Brief communication (http://www.the-cryosphere.net/about/manuscript_types.html)**

5        *We only wanted to include data for which we had the highest confidence. Other videos from the 2014-2015 field campaign were not presented because there were either not sufficient particles in transport to give meaningful average profiles, or the particle flows were too dense and obscured illumination.   We have now added new data from the 2016 field campaign – there is now more than six times the length of the original data available for analysis and has allowed more certain generalization*

10  *of the results to inform the conclusions.*
*(Page 7, line 6-25. Page 8 line 10-15.)*

-relating to the item 3

15  Neutral stability did not occur during the field campaign as it can be seen on table 1 (the Reynolds stress is not constant with the height). It is a pity because neutral con- ditions are quite usual on the site (80% of the 192 hourly periods studied in Helgason and Pomeroy, 2005).

        *The Helgason and Pomeroy (2005) Kananaskis valley bottom site is 14 km away and 600 m*
20  *lower than this high alpine valley site and so stability is not really comparable between them. This distinction is made clearer in the revised paper.  But it was very useful that you pointed this out. Correcting a numerical error now shows, using both Monin-Obuhkov and the bulk/gradient Richardson number approaches, that there are indeed near-neutral (slightly stable) conditions on the nights of Mar 23, Feb 3 and Mar 3. All three nights are now part of the analysis and discussion and are further*
25  *detailed in the revisions.*
*(Page 9, line 1. Supplementary Document, Figure 1-3)*

So, authors can't calculate the aerodynamic roughness which becomes a function of height.

30        *Exactly, this is one complication we wished to highlight for use of log-law models that drive blowing snow in complex terrain. The discrepancy in roughness length measurements is evident in Table 1. There are somewhat more consistent values of $z_0$ on Feb 3 and Mar 3 for some of the recordings, now included in the new Table 1 (Table 2).*
*(Page 9, line 20 to Page 11 line 7.)*

What are the Richardson numbers for these experiments? Is there any other data under neutral stability over the course of the campaign?

        *Plots of the Richardson numbers for the near-neutral conditions on Mar 23, Feb 3, and Mar 3*
40  *are supplied as well as the Min, and Max values during the periods of active blowing snow recording (highlighted blocks in the time series). Slightly stable conditions occurred on all nights, and this correction is included in the revised paper. Monin-Obukhov length also indicates slightly stable*

*conditions during all recording periods.*
*(Supplementary Document, Figure 1-3)*

For recording 2 there is a strong difference between u* estimated by eddy covariance method at a height
of 200 cm and a height of 40 cm. Is it possible for the blowing particles to disturb the measurements?

*Yes, it appears that was the case. While the noise did not greatly affect the mean wind speed (there is reasonable agreement between 40 cm and 200 cm averages), there was sufficient noise in the signal to obscure the turbulence measurements. This is noted in the revised paper.*
*(Page 9 line 17).*

What are the drifting snow fluxes measured during recording-only period and during the 15 minutes surrounding each recording.

*Equivalent diameters of individual blowing snow particles were measured in each frame, allowing a spherical estimate of the blowing snow volume in the 2mm wide plane of illumination. This is similar to the sand and snow studies of Creysells et al. (2009), Guo et al. (2013), and Paterna et al. (2016) among others. The time series of volume fraction was then multiplied by the density of ice by the average particle velocity for each time step to obtain a mass flux of blowing snow $Q_s$ in kg $m^{-2}s^{-1}$. Averaging these values over the duration of a recording gives a mean blowing snow flux rate for given wind and snow conditions. Values of $Q_s$ are now included in the wind characteristics Table 2.*
*(Page 8 line 4-9)*

There are quite unusual results which need to be commented (for example high value of roughness which can be smaller when estimating by flux-profile estimation techniques suggesting that the mean wind profile was in equilibrium with the snow surface, however the turbulence was not (Helgason and Pomeroy, 2005).

*We do not attempt to estimate roughness from the wind speed profile given the violation of fully developed log-linear law assumptions. This is quite different from H and P's situation where time-averaged log-linear profiles appeared to be valid. However the high roughness lengths due to high turbulence appear to be characteristics of both sites due to their mountain location. This is commented on further in the revised paper.*
*(Section 3.1)*

**- (4) The text could be more concise and focused. Similar points are discussed in several places of the text.**
-relating to the item 4

P7-L10/20 : Paragraph about Ascending particles
P7-L21/P8-L8 : Paragraph about Descending particles
P8-L9/P8-L14: Paragraph about Ascending particles. It is a little bit confusing for the reader-

*Thank you for this excellent suggestion. These sections have been reorganized.*

**- (5) Moreover the nature of discussion should be more quantitative than qualitative.**
relating to the item 5

Figure 5 by itself is not an evidence that tumblons eroded many smaller crystals from the surface or shattered themselves and immediately became saltating grains, depending on impact velocity. Where are the measurements to show the effect of impact velocity? Moreover impacting particles may travel transverse to the plane of light and may not be included on the second image. Conclusions must be based on a statistical approach.

*Tumblons have not been identified before in two-phase flow and their initial description here is necessarily mainly qualitative. The point of their inclusion in the paper is to distinguish the types of motion found in natural, outdoor blowing snow from that found in sand or found in wind tunnels. Figure 5 has been removed and the reference has been changed to supplemental videos, with overlain PTV vector fields, of tumblons that both shatter upon impact and tumble along the surface with. To better quantify them, we now compare the impact velocity of the shattering tumblon and that which remained whole.*

*Shattering Tumblon:* $ \approx < 2.46, -.43 >$ $ms^{-1}$
*Rolling tumblon:* $ \approx < 0.6, .1 > ms^{-1}$ *(depending on what is considered center of mass).*

**- (6) Papers supporting the reasoning should be properly referenced and used. Otherwise, it puts a doubt into readers' minds.**
-relating to the item 6
P3- L7: Ho et al., 2011 does not address grain velocity distribution functions

*Thank you - corrected.*

P8-L27: Ho et al., 2012 deals with Particle velocity distribution in saltation transport. So when speaking about number density, the authors have to use the right reference (It is probably Ho et al., 2011).

*Thank you - corrected.*

Ho et al., 2011 explained that the particle volume fraction decreases with height at a given exponential rate in saltation layer. If the authors want to compare their results with Ho et al., 2011, they have to limit the analysis to the first centimeter and to draw the result in the same manner as Ho et al., 2011 (figure 8) with an inset including the characteristic decay length.

*This has been done for 12 videos over the nights of Mar 23 2015, and Feb 3 and Mar 3 2016. Because our study is concerned with highly intermittent conditions, we profiled average particle*

*number flux (number of particles in transport multiplied by average particle velocity at that height). With 100% tracking, in equilibrium wind tunnel conditions, this value is theoretically equal at every time step. As our time series include periods of developing and no flux, a temporal average of particle fraction would not be comparable to the results of Ho et al. (2011). Instead, we limit our attention to only periods when transport was occurring. Notably, this means the characteristic decay lengths are not "apples to apples" comparable with Ho et al. 2011, but the fraction of total flux at each height in our profiles is representative of a plot that could be generated by Ho et al. (2011) data. Analysis of changes between our decay lengths in the style of Ho et al. (2011) is possible for our data set has been conducted.*

*This is discussed in more detail in the revised paper.*
*(Figure 7, Page 14, line 3-21. Page 21, last paragraph)*

Moreover the authors base their analysis on the fractional particle number flux whereas Ho et al., 2011 base their analysis on the particle volume fraction. If both results are compared, the authors have to take into account the volume of particles which can vary according to the wind speed.

*The methods of obtaining the two data sets, and the values under comparison are now more clearly distinguished. The difference between mass flux (or volume fraction) and number flux has been mentioned in the text with respect to the ability of the tracking algorithm to capture the flux: "As noted elsewhere (Creyssels et al., 2009), particle-tracking algorithms in the densest regions of saltation are still problematic, and thus these are conservative underestimates of near-surface flux." –page 9.*

*This is discussed in more detail in the revised paper:*
*(Page 14, last paragraph, Page 15, line 3-9, Page 21 last paragraph)*

P2-L29: Schmidt (1980) instead of Schmidt (1986)

*Thank you - corrected.*

P11-L18/24: Sugiura and Maeno, 2000 made a distinction between horizontal restitution coefficient and vertical coefficient restitution, which are different from the restitution coefficient calculated from the authors. Moreover the calculation method completely differs. The authors should make it clear.

*We have now noted that we used a different method, and that because of the density of flow a statistical method was necessary: A particle by particle restitution was not possible because the tracking software used often re-identified a rebounded particle as distinct from an incoming particle. Therefore a bulk statistical approach was used, what was the average incoming versus the average outgoing velocity. Looking in a region >1 particle diameter above the surface, we hoped to measure a rebound coefficient for truly rebounding particles, and not including reptating grains, which would artificially lower the coefficient. In the revised paper, we have lifted the region to 6 mm ($\pm 2\,mm$) in terrain following coordinates above the surface.*

*Additionally, because we are dealing with a natural surface topography, not a smooth bed as found in wind tunnel studies, parsing out horizontal and vertical restitution coefficients is not beneficial as the impact angle depends largely on the location of impact. A more general approach of E_xy does not require distinguishing the location of impact and provides information on kinetic redistribution and grain-bed impact elasticity useful for momentum balance concerns.*
*(Page 16, First Paragaph)*

P7-L19: What is the numerical value of the transition height obtained by Ho et al., 2014 (2zf ) ? As far I can see from Figure 3 the Bagnold focus point zf is around 8 mm. Considering uncertainties in relation to the choice of 10 mm threshold both values are close together.

*The Bagnold focus point is unavailable for us to identify, as the wind profile did not remotely follow a log-linear scale as would be found for a prairie sites (Pomeroy and Gray, 1990). This is expected given the non-steady state conditions. Therefore, a log-linear profile with a linear offset (addition of U_f term) does not fit either. Attempts to fit it with and without the Bagnold focal point produced focal point values ranging from 11 mm to nearly 6 m!*
*(Page 10, line 5)*

What are the Shield numbers of the snow particles in the experiments? Ho et al., 2014 remain that the results have been obtained in a finite range of Shields number from 0.04 to 0.2.

*Shields numbers ranged from 0.006 to 0.2. This is now indicated in the wind characteristics table 2.*
*(Page 12, line 23, Page 13, line 2.)*

Bauer, B. ., J. Yi, S. Namikas, and D. Sherman (1998), Event detection and conditional averaging in unsteady aeolian systems, *J. Arid Environ.*, *39*, 345–375.

Creyssels, M., P. Dupont, a. O. El Moctar, A. Valance, I. Cantat, J. T. Jenkins, J. M. Pasini, and K. R. Rasmussen (2009), Saltating particles in a turbulent boundary layer: experiment and theory, *J. Fluid Mech.*, *625*, 47–74, doi:10.1017/S0022112008005491.

Guo, L., and N. Huang (2013), Wind tunnel studies on the vertical emission of sand grains from surface, in *AIP Proceedings 1542*, vol. 1087, pp. 1087–1089.

Helgason, W., and J. Pomeroy (2005), Uncertainties in estimating turbulent fluxes to melting snow in a mountain clearing, in *Proc. 62nd Eastern Snow Conf*, pp. 129–142.

Ho, T. D., A. Valance, P. Dupont, and A. Ould El Moctar (2011), Scaling laws in aeolian sand transport, *Phys. Rev. Lett.*, *106*(9), 4–7, doi:10.1103/PhysRevLett.106.094501

Ho, T. D., P. Dupont, A. Ould El Moctar, and A. Valance (2012), Particle velocity distribution in saltation transport, *Phys. Rev. E - Stat. Nonlinear, Soft Matter Phys.*, *85*(5), 1–5, doi:10.1103/PhysRevE.85.052301.

Ho, T. D., A. Valance, P. Dupont, and A. Ould El Moctar (2014), Aeolian sand transport: Length and height distributions of saltation trajectories, *Aeolian Res.*, *12*, 65–74, doi:10.1016/j.aeolia.2013.11.004.

Paterna, E., P. Crivelli, and M. Lehning (2016), Decoupling of mass flux and turbulent wind fluctuations in drifting snow, *Geophys. Res. Lett.*, 1–7, doi:10.1002/2016GL068171.

Pomeroy, J., and D. Gray (1990), Saltation of snow, *Water Resour. Res.*, *26*(7), 1583–1594.

Schönfeldt, H.-J., and S. von Löwis (2003), Turbulence-driven saltation in the atmospheric surface layer, *Meteorol. Zeitschrift*, *12*(5), 257–268, doi:10.1127/0941-2948/2003/0012-0257.

[revised manuscript text omitted]

Figure 4 shows varying Reynolds Stress ($RS = u'w'$) generation during each recording following the

15    language of quadrant hole analysis (*Willmarth and Lu*, 1971). Sweep and ejection events (Q2 and Q4) often contributed the majority of RS at both anemometers, with a more pronounced role closer to the ground, indicating changes in the snow surface influence on wind mechanics. Q2 and Q4 stress also occupied a disproportionately small amount of time near the surface, as can be seen in the impact factors inset in the bar graphs ($IF = (\% \, Reynolds \, Stress)/(\% \, Time)$) that are greater than unity.

20    Therefore, when strong events are captured during the recordings RS values can be much larger than long time averages. The presence of a single pronounced sweep event in Recording #3, March 23 (discussed more in Section 3.3) contributed to a high turbulence intensity (45%), and a much higher friction velocity than the fifteen-minute values (0.48 m/s and 0.24 m/s respectively) and will be discussed in detail in Section 3.3.

[revised manuscript text omitted]
. Using a height separation of 3 mm for three low wind speed recordings (23 March #2 and 3 March #1 & 4) and 4 mm for all others, Fig. 8 displays horizontal velocity histograms for descending snow grains above and below these designations. For every night there was a denser surface flow whose mean statistics are dominated by slow moving particles. This is to be expected from the particle velocity and number flux profiles in Fig. 6 and 7. Transitioning away from the densest flow, the upper region histograms show saltating particles have already begun to transition towards a higher energy population, with transport dynamics centred around larger horizontal velocities resulting in a higher mean. The increasing proportion of high-energy particles with distance from the surface was due to the need for a greater velocity to reach greater heights on a ballistic trajectory from the surface and the subjection to stronger winds with increasing distance above the surface.

What is of most interest here is that a separating height was chosen such that for each recording the two histograms begin to agree for the highest energy grains (Fig. 8). That is, the highest energy population of descending particles present in the upper band can also be identified in the surface flow and that high

**Nik Aksamit 8/31/2016 2:41 PM**

and low-energy populations coexisted as part of a continuous spectrum of motion at the surface. This was indicative of an inherent coupling of the creep and saltating grains. Working with the assumption that the surface histograms contain contributions from both the high and low-energy populations of grains, whereas the upper band histogram consists only of high-energy grains, we can then identify the low-energy population at the surface as the difference of the two collections of particle vectors. With this, we can begin to quantify the role that creep plays in surface impact and wind momentum balance. Note, identifying creep particles is only possible when using both the particle velocity and location because high-energy grains were also impacting surface. The difference of surface and upper region populations showed a low-energy near-surface population that constituted 44-52%, 21-29% and 37-61% of the number flux on March 23, February 3, and March 3 respectively. Thus the slow moving low-energy creep grains were a considerable contribution to total particle flux below 20 mm over all snow types and wind conditions observed.

In the near-surface region, the ability of the snowpack to redistribute impact momentum was estimated from PTV data derived from each recording. Snow particle rebound efficiency varied from night to night, and was quantified by the restitution coefficient, $\overline{e_{xz}} = \overline{\|s_r\|}/\overline{\|s_t\|}$, where $\overline{\|s_r\|}$ and $\overline{\|s_t\|}$ are mean ejection and impact speeds of particles, respectively, at 6 mm $\pm$ 2 mm such that the lower bound of the measurement band corresponds with the upper bound of the surface band generating Fig. 8 histograms. Because of the density of the particle flow, and transverse components of travel, a bulk statistical approach must be used to quantify momentum redistribution into the particle bed. Therefore particle ejection speeds of both the rebounding grains and the splashed grains that reach ~20 particle diameters above the surface were averaged. Over the course of the campaign, $\overline{e_{xz}}$ varied from 0.58 to 0.84, within the bounds of the previous wind tunnel blowing snow study of *Sugiura and Maeno* (2000) who used a complimentary particle by particle approach, separating individual rebounding and splashed grains. The mean restitution coefficient was 0.69 for the graupel grains on March 23, 0.79 for fresh snow on March 3, and 0.73 for old snow on February 3. This suggests rebound efficiency was dependent on time-sensitive saltating snow crystal and bed mechanical/material properties, as also noted in a blowing snow wind tunnel study by *McElwaine et al.* (2004).

Using $\overline{e_{xz}}$ from each recording and assuming a constant mean grain diameter, it was then possible to

sum the total impacting high-energy snow grain momentum and to estimate how much momentum was lost to the snow bed through surface bond breaking, particle shattering or particle ejection. Similarly, one can determine the amount of momentum present in the low-energy surface population. Consistently, there was a large residual in the momentum balance, with up to 10 times the momentum in the low-energy surface particles than that which was available from particle impacts. This ratio of the sum of momentum in the creeping population to that lost to the surface from impacting high-energy particles capable of splash is indicated on each recording histogram in Fig. 8 as "Crp/Spl". All coefficients greater than one indicate a creep layer with total momentum greater than that which was possibly gained through saltating particle impacts. The creep layer was not merely a region of splashed grains that balances the momentum lost from saltating particles, but also behaves as a sink of wind momentum as well, further complicating the lower boundary layer condition for wind models.

**3.3 Turbulent Event Transport**

The initiation mechanisms observed at the surface during the onset of transport events differ from those proposed in single threshold velocity models (e.g. *Schmidt*, 1980), suggesting multiple thresholds with the dense surface flow playing a crucial role. All three transport thresholds recognized during video playback are crossed during 23 March recording #3 (Fig. 9) where an isolated gust was captured with minimal antecedent transport. Thus, it will be used as an example for further discussion. Concurrent streamwise wind measurements at 200 and 40 cm are plotted in Fig. 9a showing penetration of a turbulent sweep to the surface that is responsible for snow transport. In Fig. 9a, filled circles indicate sweep events with RS exceeding one standard deviation of total RS (colors corresponding to measurement heights), while triangles indicate similar moments of strong ejections. Fig. 9b and 9c show time series of spatially averaged particle velocities, and total particles tracked, respectively, in three height bands, $1 < z < 4\ mm$ (Near-Surface), $4 \leq z < 8\ mm$ (Mid), and $8 \leq z < 30\ mm$ (High). These three heights were chosen to demonstrate the subtle differences in particle transport and the continuum of motion as grains began motion and began bouncing to greater heights as wind speeds increased. These are not hard thresholds of "creep" versus "saltation" regimes. Fig. 9d shows the time series of instantaneous blowing snow flux $Q_s$ in $kg\ m^{-2}s^{-1}$. These binarization based flux

Nik Aksamit 8/31/2016 2:41 PM

Nik Aksamit 8/31/2016 2:41 PM

Nik Aksamit 8/31/2016 2:41 PM

Nik Aksamit 8/31/2016 2:41 PM

Nik Aksamit 8/31/2016 2:41 PM

Nik Aksamit 8/31/2016 2:41 PM

Nik Aksamit 8/31/2016 2:41 PM

Nik Aksamit 8/31/2016 2:41 PM

[revised manuscript text omitted]
 wind accelerates, more particles are readily transported to upper heights and accelerated, vertically distributing the mass flux more uniformly. Increasing $h_v$ values with $u_*$ support a decreased role of creep in the near-surface mechanics. This occurs gradually with increasing wind speed, rather than involving discrete transport threshold velocity values for separate modes of transport. The notable exception for this trend in $h_v$ was the night of March 23 (Fig. 7, inset – red dots) where $h_{v,}$ $l_{v,}$ and $v_0$ values remained constant as was predicted by *Ho et al.* (2011) for erodible beds. This difference in behaviour can be physically justified as for spherical graupel grains over a non-cohesive bed, the conditions most closely resembled sand, and so the number flux profiles behaved more similarly to sand than for the other two nights of blowing snow.

Histograms of horizontal velocity further supported the relevant and dynamic role of the low-energy surface population of blowing snow (Fig. 8), and the decreasing importance of near-surface mass flux with increasing surface hardness. Uniformly, there is an imbalance of momentum lost by impacting high-energy grains and that found in the creep layer. This created a requisite wind to creep momentum transfer that is much more substantial over erodible beds. The ratio of particle momentum in creep to that which was contributed by splash is much higher over erodible beds with lower HHI (March 23 and March 3) than on the wind hardened rigid bed night of Februrary 3 (4-10 versus 2, Fig. 8). This complimented the fact that the low-energy surface transport on February 3 had the smallest contribution to total number flux. Thus particle type and snow bed properties played a significant role in the surface momentum balance, changing the uniformity of saltation profiles and wind momentum lost directly to creep. For the erodible bed recordings (March 23, 2015 and March 3, 2016), the momentum present in the slow surface transport was often larger than that found in the upper region high-energy grains as can be seen by Crp/Spl > 4. For events with larger grains and lower surface hardness, transport initiation dynamics changed as wind played a larger role than splash in generating creeping particles. 
[revised manuscript text omitted]

---

## Referee Report (RR1)

Report on

Near surface snow particle dynamics from particle tracking velocimetry and turbulence measurements during Alpine blowing snow storms

By N.O Aksamit and J. W. Pomeroy

The reviewer really appreciates efforts made by the authors to improve the manuscript by adding new references, new  graphs  and data allowing generalization of the results. The main points raised have been answered.

Discussions concerning wind characteristics and more particularly the quadrant hole analysis are a prime example: conclusions are now convincing but lead to new comments. Why did the authors choose a fifteen minute average to calculate the mean wind speed and the friction velocity and what influence does this choice have on the quadrant hole analysis (Figure 4).

My main other comments are still in regard to the limit between creep and saltation. The authors cannot hide behind the arguments: "the arbitrary height is not a threshold, it was chosen to illuminate the subtle difference in particle transport and the continuum of motion…". This argument can be admitted for 3.3 Turbulent flow and Figures 9.  But in order to calculate the ratio of the sum of momentum in the creeping population to that lost to the surface from impacting high-energy particles capable of splash, the authors need to be very precise as to the method used and to explain as clearly and in as much details as possible the reasons leading to the values 3 and 4 mm. In such a way, similar experiments with different hardness can be conducted in the future and compared with the results presented in this paper. P28 it is written that this choice is made according the high and low-energy saltating grain populations theory proposed by Ho et al, 2014. I would like to recall the conclusions drawn by Ho et al., 2014:

As a result, two different populations can be identified but using a criterion that differs from that of Bagnold:

    (i)       the low energetic saltating particles lying below $2z_f$ and hardly influenced by change of the flow strength
    (ii)      the highly energetic ones saltating above $2z_f$  sensitive to increase of the flow intensity.

On what figures or $z_f$ values do the authors base the choice of the threshold 3 or 4 mm ?

Moreover  the method used for calculating Crp/Spl is a little ambiguous for the reader and need to be clarified. The low-energy population at the surface is identified as the difference of the two collections of particles vector (P 29 line 5). As far I understand, this implies that the high-energy population (particles capable of splash) is identified to be twice of particles vector for upper region height bands. If not the total impacting high-energy snow grain momentum will be wrong. But later (p35 line 24) the authors wrote « the momentum present in the slow surface transport was often larger than that found **in the upper region high energy grains** as can be seen by crp/Spl>4." It is really confusing because it seems in this sentence that only particles in the upper region have been taken into account. For clarity it will be helpful to give an example showing a horizontal particle velocity histogram for near surface and upper regions and the corresponding histogram for creeping population and particles capable of splash.

A few minor errors have been made:

Figure 6 d-f) Friction velocity versus particle velocity gradient
Figure 6 g-i) Friction velocity versus particle slip velocity
Figure 9-b) Mean horizontal particle velocity for three height band
And some graphs are still not enough clear.

P32 line 4 / At 9 s, the number of particles tracked in the upper regions increased as particles tracked near the surface decreased. From my point of view it is not visible in the graph. It will be probably useful to zoom in on the concerned area.

---

## Author Response (AR2)

Reviewer 1:

In general, the manuscript has been largely improved with including the new data and expanding the data analysis part; total volume is nearly the twice as the previous one at this stage. On the other hand, I have got a feeling that it became too wordy. I hope the authors eliminate the repetitions and the redundant part, and make the manuscript more compact.

*The authors have compacted the writing style in all sections, and removed repetition that arose between results-discussion-conclusion sections. However, following suggestions from this and the other reviewer, explanations have been expanded in several sections.*

Further, I am still concerned about the surface topography. In fact, 5 mm rise over 130 mm is not enormous and is inevitable in nature, I understand. However, as you may see, particle motions are very sensitive on the surface feature. Thus, I am afraid that the particle speed in Fig. 6 and the horizontal flux in Fig.7 are strongly affected, even though the terrain-following coordinates are adapted. Such as, the particle located at x=60 mm and z=10 mm has a history which passed over the snow surface at x=0 mm where the snow surface is 5 mm lower. It should be carefully taken into account in the analysis.

*We agree, surface topography is incredibly difficult to account for, especially with the limited frame height available in these recordings. A similar example to what you are describing was described on page 6, line 25 in this version. We have further detailed the advantages terrain following coordinates, and issues with near surface flux profiles in nature. (page 6 line 15 – page 7 line 7)*

Although it is not shown in revised version, in Fig. 3 (g) of the previous manuscript, particle flux showed the maximum at 2 to 7 mm high. This is presumably caused by the effect of topography change.

*Yes. We have expanded the explanation of terrain following coordinates on Page 6 to include mention of the 7 mm height flux maximum in immersed boundary coordinates and the difficulty identifying locations of particle flux maximums in natural conditions.*

Bagnold (1941) used the term 'creep' to describe grains rolling and jostling along the surface. Thus I do have slightly an aversion to call the particle motion less than 4mm as creep. Probably "reputation" that means "move slowly" introduced by Ungar and Haff (1987) is more suitable. At any rate, 4 mm is several tens times larger than the mean particle diameter and, needless to say, large number of saltation particles are involved there.

*We agree with this statement and have revised the literature, especially figure 8 and the section containing Creep momentum balances to explicitly detail that the particles below 2-5 mm (now replacing 4 mm threshold) are comprised of both high and low-energy populations. Those slow moving grains that exist solely below these heights are referred to collectively as near-surface grains or surface residuals, noting that creep is contained*

*in this area. The concept of creep is still used to connect with the existing literature. Ungar and Haff (1987) use reptating and creeping grains interchangeably (page 294, 1ˢᵗ paragraph).*

Further, although it is not related to the evaluation of the contents directly, the authors need to reply to the reviewer's comment line by line. 'Refer to the response to other reviewer' is not an appropriate attitude.

*This will not happen again.*

As a whole, I do believe this manuscript is approaching to the publishing stage. However, I am glad if the authors can clarify and improve some of the points listed below before the publication.
Specific comments are listed below.

Page 4, line 23: I do not see where the potential for adapting the continuum model to the blowing snow phenomena is discussed in the manuscript.

*The combination of all results found in this manuscript support the improvement of the physical accuracy of blowing snow models if a continuum model of transport is adopted, with preliminary trends relating continuous velocity and concentration profiles with wind conditions (Section 3.2 and Discussion). Thus with further observations, adapting a continuum model could be possible and beneficial: Page 12 line 21, Page 15 1ˢᵗ and 2ⁿᵈ paragraph, Page 18 line 17, Page 22 line 5, Page 23 last paragraph, Page 26 1ˢᵗ paragraph.*

Figure 5: (a) to (c) are not shown in the figure. The scale of the vertical of upper two figures needs to be set equal.

*Thank you. The figures have been rescaled. The labels (a) to (c) were previously in the bottom left of the plots and have been made more obvious.*

Page 9, line 4: We cannot expect the steady-wind condition in nature, except for the specific situation like in Antarctica.

*Agreed. This may be a simple statement, but the authors believe this fundamental sentiment is currently lacking in the blowing snow literature, especially when modelers attempt to adapt steady state equations to complex terrain. This is also why we used the less-strict steady state test of Foken and Wichura (1996), which can be satisfied in nature, such as throughout the FLUXNET network (Baldocci et al. 2001).*

Page 10, line 6: Why don't you say "the focal point was not recognized in this study"?

*This has been adapted. Thank you.*

Figure 6: Please check the figure caption. d) - f) are for the particle velocity gradient and

g) – i) are for the particle slip velocity.

*Done. Thank you.*

Page 15, line 18: I don't see how you can expect the dense surface flow with profiles in Figs. 6 and 7. The particle number flux increases with approaching the bed surface, but no specific layers with maximum flux are recognized. In actual, as pointed previously, in Fig. 3 (g) of the previous manuscript, particle flux showed the maximum at 2 to 7 mm high. However, this is presumably induced by the topography change.

*The maximum is at a height of zero for all plots in Figure 7 (which includes the previous Figure 3(g) on a log-linear plot with y-values being terrain following coordinates) and is the maximum for the fitted equation of fractional number flux (page 14 line 5). The effect of topography on concentration peak was noted in both the previous manuscript as "microtopography" in Figure 3g, page 4 line 2, page 6 line 27, and page 8 line 2. This has been addressed in the present manuscript in the justification of terrain following coordinates page 6 line 15 – page 7 line 7. The terrain following coordinates have essentially removed the mellow gradient from left to right seen previously, as well as small peaks, though with changes in particle orientatin. Additionally, the $F_z$ plots are now complimented with the proportion of total momentum found in the surface residual, the dense surface flow that does not rise above 2-5 mm, page 16 2$^{nd}$ paragraph.*

Figure 8: In the figure caption, no explanation that this is for just "descending particles" is shown.

*Corrected. Thank you.*

Page 17, line 11: Shear stress due to the wind can be estimated from the ultra sonic anemometer data. Thus, it will be informative to compare with the particle momentum near the surface quantitatively.

*Friction velocity values have been calculated using shear stress estimates from the ultra sonic data, thus they are an analog to the mean shear stress (often referred to as Reynolds stress to indicate use of u'w'). A discussion has been added comparing shear stress and the amount of momentum present in the "surface residual." A comparison of friction velocity and surface residual momentum can be found on Page 22 last paragraph, bottom left - figure 8.*

Page 17, line 23: The definition of three height bands should be defined earlier, when the data of near-surface and high bands were introduced.

*The three height bands are independent of the two-region near-surface and upper-region momentum analysis earlier in the text. As well, the previous "upper and lower region" has been modified to a moving threshold instead of the strict 3 or 4 mm used previously, and thus does not line up with the three height bands.*

Page 24, line 27: Descriptions of the article number concentrations will be useful to discuss whether it will be comparable with the ordinary fluidized-bed and be applicable the physical properties of the bed.

*Particle number concentrations directly from PTV would be misrepresentative because particles are only counted if they can be accurately tracked. This is why we stayed consistent with representing profiles as percentages of flux of particle counts (Section 3.2). As noted elsewhere (Creyssels et al., 2009), obtaining perfect tracking at the surface is currently nearly impossible. Discussions about current improvement in PTV software have indicated that analysis techniques are being developed to improve PTV in incredibly dense flows. Other avenues of individual particle counts in dense surface flows are also being pursued for future research.*

Baldocchi, D. et al. (2001), FLUXNET: A new tool to study the temporal and spatial variability of ecosystem-scale carbon dioxide, water vapor, and energy flux densities, *Bull. Am. Meteorol. Soc.*, *82*(11), 2415–2434, doi:10.1175/1520-0477(2001)082<2415:FANTTS>2.3.CO;2.

Ungar, J. E., and P. K. Haff (1987), Steady state saltation in air, *Sedimentology*, *34*(2), 289–299, doi:10.1111/j.1365-3091.1987.tb00778.x.

Reviewer 2:

The reviewer really appreciates efforts made by the authors to improve the manuscript by adding new references, new graphs and data allowing generalization of the results. The main points raised have been answered.

Discussions concerning wind characteristics and more particularly the quadrant hole analysis are a prime example: conclusions are now convincing but lead to new comments. Why did the authors choose a fifteen-minute average to calculate the mean wind speed and the friction velocity and what influence does this choice have on the quadrant hole analysis (Figure 4).

*Both fifteen minute and recording specific values are included for turbulence characteristics surrounding all recordings in table 2. To be clear, Figure 4 was not generated using 15-minute values, and this is further emphasized page 10 line 18. The difference between 15-minute and recording specific turbulence characteristics is outlined on page 10 1ˢᵗ paragraph, and page 20 1ˢᵗ paragraph, with new expansion on turbulence intensity. The discussion of 15-minute values in the previous paragraph was used to discuss the nature of the wind over the entire nights of recording, and comparing to a nearby study site.*

My main other comments are still in regard to the limit between creep and saltation. The authors cannot hide behind the arguments: "the arbitrary height is not a threshold, it was chosen to illuminate the subtle difference in particle transport and the continuum of motion…". This argument can be admitted for 3.3 Turbulent flow and Figures 9.

*Thank you. This comment is limited to Section 3.3 and Figure 9. Page 17, line 27.*

But in order to calculate the ratio of the sum of momentum in the creeping population to that lost to the surface from impacting high-energy particles capable of splash, the authors need to be very precise as to the method used and to explain as clearly and in as much details as possible the reasons leading to the values 3 and 4 mm.

*The surface momentum has been further analyzed in the manuscript. Values of 3 and 4 mm implicitly defined an upper boundary of creep, and this was not the intention of the authors. The comparison between a "surface residual" and upper-region grain momentum has been introduced, with varying heights from 2 – 5 mm based on the capabilities of the camera and PTV software (page 15 line 25, page 16 line 10). We no longer calculate a creep momentum.*

In such a way, similar experiments with different hardness can be conducted in the future and compared with the results presented in this paper. P28 it is written that this choice is made according the high and low-energy saltating grain populations theory proposed by Ho et al, 2014.

*It has been clarified that the separation at varying heights supports the concept of two*

*populations existing at the near surface region, and is not driven by Ho's methods. We do not use $z_f$ (page 10 line 10, page 15 line 16).*

I would like to recall the conclusions drawn by Ho et al., 2014:

As a result, two different populations can be identified but using a criterion that differs from that of Bagnold:
(i) the low energetic saltating particles lying below 2zf and hardly influenced by change of the flow strength
(ii) the highly energetic ones saltating above 2zf sensitive to increase of the flow intensity.

*The similarities and difference of response of low-energy surface grains to changes in flow strength with respect to Ho et al. 2014 are mentioned in the discussion. Page 21 1st paragraph, page 22 2nd paragraph.*

On what figures or zf values do the authors base the choice of the threshold 3 or 4 mm ?

*This has been changed to 2-5 mm and is explained to be limitations of the PTV and camera setup (page 10 line 10, page 15 line 16)*

Moreover the method used for calculating Crp/Spl is a little ambiguous for the reader and need to be clarified. The low-energy population at the surface is identified as the difference of the two collections of particles vector (P 29 line 5). As far I understand, this implies that the high-energy population (particles capable of splash) is identified to be twice of particles vector for upper region height bands. If not the total impacting high-energy snow grain momentum will be wrong.
But later (p35 line 24) the authors wrote « the momentum present in the slow surface transport was often larger than that found in the upper region high energy grains as can be seen by crp/Spl>4." It is really confusing because it seems in this sentence that only particles in the upper region have been taken into account.

For clarity it will be helpful to give an example showing a horizontal particle velocity histogram for near surface and upper regions and the corresponding histogram for creeping population and particles capable of splash.

*This has explanation has been expanded (Page 15 line 15 to Page 16 line 25). Moreover, the suggested histogram has been added to Figure 8.*

[revised manuscript text omitted]